# E3 ligase Nedd4l promotes antiviral innate immunity by catalyzing K29-linked cysteine ubiquitination of TRAF3

Peng Gao [1,7], Xianwei Ma [2,7], Ming Yuan [3,7], Yulan Yi [1,7], Guoke Liu [1], Mingyue Wen [3], Wei Jiang[1], Ruihua Ji [4], Lingxi Zhu [1], Zhen Tang [1], Qingzhuo Yu [1], Jing Xu [1], Rui Yang[4], Sheng Xia [5], Mingjin Yang [3], Jianping Pan [6], Hongbin Yuan [4✉] & Huazhang An [1✉]

Ubiquitination is one of the most prevalent protein posttranslational modifications. Here, we show that E3 ligase Nedd4l positively regulates antiviral immunity by catalyzing K29-linked cysteine ubiquitination of TRAF3. Deficiency of Nedd4l significantly impairs type I interferon and proinflammatory cytokine production induced by virus infection both in vitro and in vivo. Nedd4l deficiency inhibits virus-induced ubiquitination of TRAF3, the binding between TRAF3 and TBK1, and subsequent phosphorylation of TBK1 and IRF3. Nedd4l directly interacts with TRAF3 and catalyzes K29-linked ubiquitination of Cys56 and Cys124, two cysteines that constitute zinc fingers, resulting in enhanced association between TRAF3 and E3 ligases, cIAP1/2 and HECTD3, and also increased K48/K63-linked ubiquitination of TRAF3. Mutation of Cys56 and Cys124 diminishes Nedd4l-catalyzed K29-linked ubiquitination, but enhances association between TRAF3 and the E3 ligases, supporting Nedd4l promotes type I interferon production in response to virus by catalyzing ubiquitination of the cysteines in TRAF3.

[1] Clinical Cancer Institute, Center for Translational Medicine, Second Military Medical University, Shanghai 200433, China. [2] Scientific Research Center, Shanghai Public Health Clinical Center, Fudan University, Shanghai 201508, China. [3] Immunology Department & National Key Laboratory of Medical Immunology, Second Military Medical University, Shanghai 200433, China. [4] Department of Anesthesiology, Changzheng Hospital, Second Military Medical University, Shanghai 200003, China. [5] Department of Immunology, School of Medicine, Jiangsu University, Zhenjiang, Jiangsu 212013, China. [6] Department of Clinical Medicine, Zhejiang University City College School of Medicine, Hangzhou 310015, China. [7] These authors contributed equally: Peng Gao, Xianwei Ma, Ming Yuan, Yulan Yi. ✉email: jfjczyy@163.com; anhz@immunol.org

Ubiquitination is one of the most prevalent protein post-translational modifications. Ubiquitin often conjugates to lysine residue of substrate protein, and less frequently to the N-terminus of substrate protein[1,2]. However, ubiquitin can also conjugates to non-lysine residues, such as cysteine (Cys, C)[3]. Unlike ubiquitination of lysine residues, which has been well demonstrated to play important roles in diverse cellular and biological processes, little has been known about the physiological significance of non-lysine residues being ubiquitination sites. On the other hand, each ubiquitin molecule contains seven lysine residues (K6, K11, K27, K29, K33, K48, K63). Linkage of ubiquitin to any one of the lysine residues leads to the formation of polymeric ubiquitin chains. While the ubiquitin chains formed by linkage through K48 and K11 lead the substrates to degradation, formation of K63-linked ubiquitin chains usually regulate protein interaction, activity, and protein localization. For ubiquitination in which ubiquitin chain is formed through the other lysine residues, namely Lys6, Lys27, Lys29, or Lys33, few substrates are known, and their significances are poorly understood[1,2].

Pattern recognition receptors (PRR) activate antiviral innate immunity mainly through inducing production of type I interferon[4,5]. Among the PRRs, RIG-I-like receptors (RLRs), which recognize RNA viruses, induce interferon production through MAVS (also known as IPS-1, VISA, and CARDIF)-TRAF3-TBK1-IRF3 pathway. RIG-I also mediates DNA virus-induced interferon production by sensing RNA polymerase III-transcribed RNA[6,7]. In TLR family, TLR3 and TLR4 induce interferon production through TRIF-TRAF3-TBK1-IRF3 pathway. In addition, by activating MAPK and NF-κB pathways, PRRs also induce proinflammatory cytokine production to facilitate antiviral innate immunity. Ubiquitination plays important roles in the regulation of innate immune signal transduction[8,9]. By now, the ubiquitinated sites identified in the signal molecules in innate immunity are limited to lysine residues[8,9]. It remains unknown whether ubiquitination of non-lysine residue plays a role in innate immunity signaling.

TRAF family member TRAF3 promotes type I interferon production and inhibits inflammatory cytokine production in PRR-mediated innate immunity[10–12]. Different modes of ubiquitination of TRAF3 selectively activate the expression of type I interferon and proinflammatory cytokine, with K48-linked polyubiquitination promoting TRAF3 degradation and increasing inflammatory cytokine production, K63-linked polyubiquitination promoting TRAF3 to recruit TBK1 and mediating type I interferon production[12]. The regulation of TRAF3 ubiquitination is intensively studied. E3 ubiquitin ligases, such as cIAP1/2, Peli1, and TRIAD3A catalyze K48-linked ubiquitination of TRAF3 and promote TRAF3 degradation[12–14]. Several molecules regulate K63-linked TRAF3 ubiquitination. Kinase CK1ε phosphorylates TRAF3 and thereby promotes K63-linked ubiquitination, positively regulating antiviral immune responses[15]. De-ubiquitylating enzyme DUBA negatively regulates antiviral type I interferon production by removing K63-linked ubiquitin chain in TRAF3[16]. Evidences suggest that TRAF3 is self-polyubiquitinated to mediate type I interferon production[12]. However, there are also reports that cIAP1/2 and HECTD3 catalyze K63-linked polyubiquitination of TRAF3 and increase type I interferon production[17,18]. How TRAF3 is ubiquitinated in innate immunity remains incompletely known.

Nedd4l (neural precursor cell-expressed developmentally down-regulated 4-like, also known as Nedd4-2) is a HECT E3 ubiquitin ligase of the Nedd4 family. Nedd4l regulates a number of membrane proteins, such as the epithelial and voltage-gated sodium channels[19]. In TGF-β signaling, Nedd4l down-regulates TGFβR1 and Smad2 to limit TGF-β signaling. Nedd4l is also involved in HIV budding[20]. In the present study, we demonstrate that Nedd4l catalyzes K29-linked ubiquitination of cysteine residues in TRAF3 and is critically required for antiviral innate immunity.

## Results

**Nedd4l deficiency impairs antiviral innate immunity**. To investigate whether Nedd4l plays a role in antiviral innate immunity, we initially used small interfering RNA (siRNA) specific to Nedd4l to inhibit Nedd4l expression in peritoneal macrophages and found that Nedd4l knockdown significantly decreased vesicular stomatitis virus (VSV)-induced IFN-β and TNF-α production in them (Supplementary Fig. 1a, b). To confirm the role of Nedd4l in antiviral innate immunity, we used a Nedd4l-deficient mouse model with spontaneous mutation in genome to investigate the effects of Nedd4l deficiency on virus-induced type I interferon and proinflammatory cytokine production. Primary peritoneal macrophages were isolated from control wild-type mice and Nedd4l-deficient mice, and then infected with VSV. Nedd4l deficiency impaired the production of IFN-β, IL-6, and TNF-α induced by the viruses at protein level, as detected by ELISA, and also at mRNA level, as detected by real-time quantitative polymerase chain reaction (RT-qPCR) (Fig. 1a, b). Consistently, Nedd4l deficiency inhibited mRNA expression of IFN-β, IL-6, and TNF-α induced by poly(I:C) transfection (Fig. 1c), demonstrating that Nedd4l positively regulated innate immunity mediated by cytoplastic RNA sensor. Similarly, Nedd4l deficiency inhibited IFN-β, IL-6, and TNF-α mRNA expression induced by poly(dA:dT) transfection (Supplementary Fig. 1c). Nedd4l deficiency also reduced IFN-β, IL-6, and TNF-α production and mRNA expression in macrophages stimulated with LPS and poly(I:C) (Supplementary Fig. 1d,e,f). However, Nedd4l deficiency did not affect IL-6 and TNF-α mRNA expression in macrophages stimulated with LTA (Supplementary Fig. 1g), suggesting Nedd4l selectively regulated innate immunity triggered by TLR family members.

The spontaneous Nedd4l-deficient mice display reduced body weights compared to wild-type control mice. To investigate the role of Nedd4l in antiviral innate immunity in vivo, we generated Nedd4l$^{fp/fp}$ mice (Supplementary Fig. 2a), crossed the mice with Lyz2-Cre transgene mice to delete Nedd4l in macrophages. Expression of Nedd4l in peritoneal macrophages and bone-marrow-derived macrophages was efficiently deleted (Supplementary Fig. 2b), while expression of Nedd4l in B cells and T cells from spleen and bone marrow were similar between Nedd4l$^{fp/fp}$Lyz2-Cre$^{+/+}$ mice and wild-type mice (Supplementary Fig. 2c). We assessed the frequency of immune cell populations in spleen, bone marrow, and blood of the mice, and found that the abundance of macrophages, granulocytes, B cells, and T cells was similar between Nedd4l$^{fp/fp}$Lyz2-Cre$^{+/+}$ and wild-type control Nedd4l$^{fp/fp}$Lyz2-Cre$^{-/-}$ mice (Supplementary Fig. 2d). These results suggest that Nedd4l myelogenous deficiency did not affect the development of major immune cell populations. The Nedd4l$^{fp/fp}$Lyz2-Cre$^{+/+}$ and wild-type control Nedd4l$^{fp/fp}$Lyz2-Cre$^{-/-}$ mice were infected with VSV by intraperitoneal injection. IFN-β, IL-6, and TNF-α concentrations in serum of the mice were then detected. As shown in Fig. 2a, conditional Nedd4l deficiency inhibited VSV-induced IFN-β and IL-6 production in vivo. Meanwhile, VSV replication and TCID$_{50}$ increased in spleen and liver in Nedd4l$^{fp/fp}$Lyz2-Cre$^{+/+}$ mice than in control Nedd4l$^{fp/fp}$Lyz2-Cre$^{-/-}$ mice (Fig. 2b, c). Consistently, the survival of the Nedd4l$^{fp/fp}$Lyz2-Cre$^{+/+}$ mice was much lower than the survival of control Nedd4l$^{fp/fp}$Lyz2-Cre$^{-/-}$ mice (Fig. 2d). These results demonstrate that Nedd4l positively regulates type I interferon and proinflammatory cytokine production in macrophages and is critical for antiviral innate immunity in vivo.

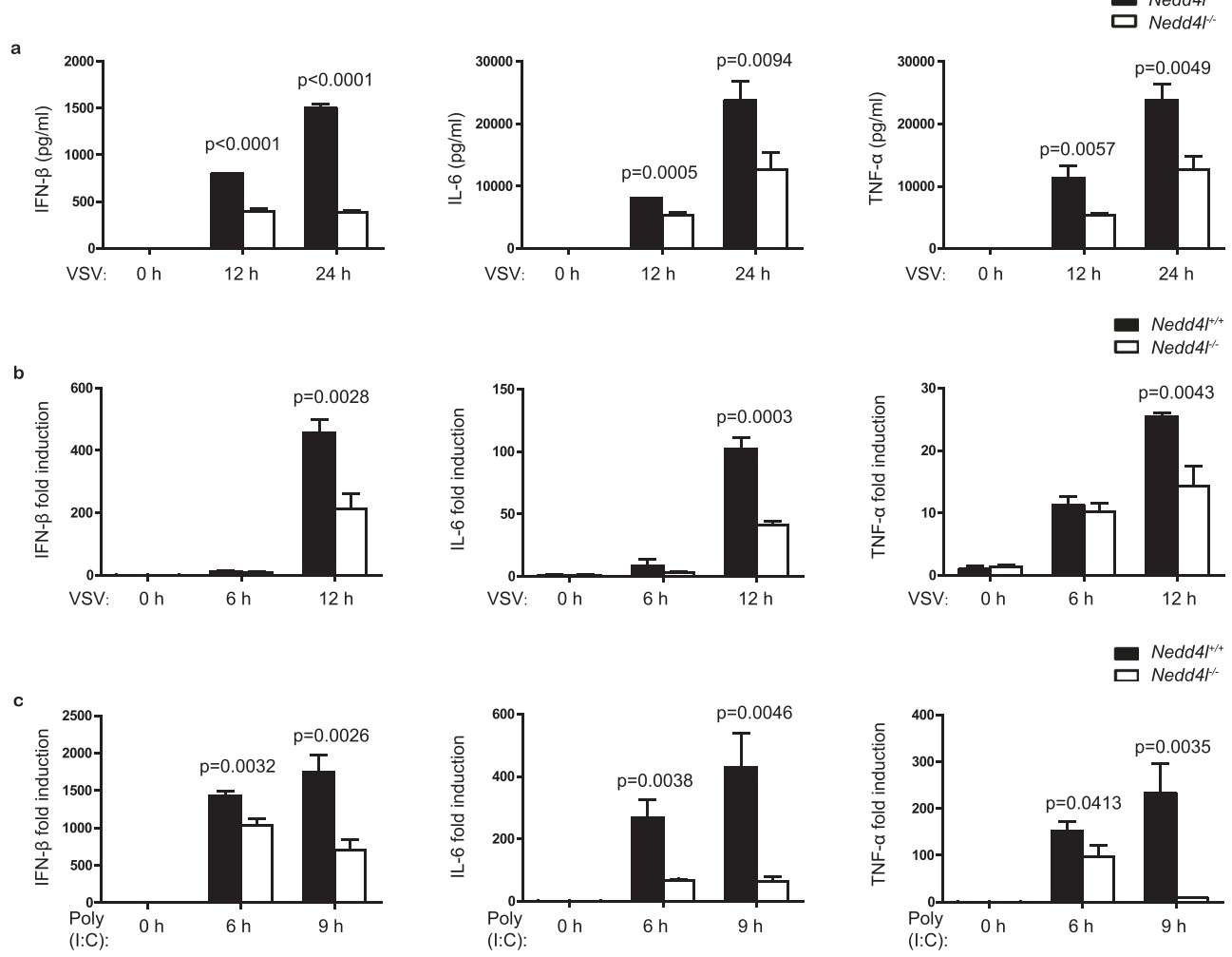

**Fig. 1 Nedd4l deficiency inhibited antiviral innate immunity in macrophages. a** ELISA assay of IFN-β, IL-6, and TNF-α production in wild-type ($Nedd4l^{+/+}$) and Nedd4l-deficient homozygous ($Nedd4l^{-/-}$) peritoneal macrophages infected with VSV (MOI = 10) for 12 and 24 h. **b** RT-qPCR analysis of IFN-β, IL-6, and TNF-α mRNA expression in peritoneal macrophages in (**a**) infected with VSV (MOI = 10) for 6 and 12 h. **c** RT-qPCR analysis of IFN-β, IL-6, and TNF-α mRNA expression in wild-type ($Nedd4l^{+/+}$) and $Nedd4l^{-/-}$ peritoneal macrophages transfected with poly(I:C) (10 μg/ml) for 6 and 9 h. Data in (**a–c**) are presented as mean ± SD ($n = 3$ per group) and $p$-values by two-tailed unpaired Student's $t$-test are indicated. Results in (**a–c**) are representative of three independent experiments.

**Nedd4l promotes TRAF3-dependent signaling**. Upon VSV infection, RLRs activate IRF3, NF-κB, and MAPKs to induce type I interferon and proinflammatory cytokine production. We detected VSV-induced phosphorylation of IRF3, NF-κB, and MAPKs in Nedd4l-deficient and wild-type control macrophages. As shown in Fig. 3a, Nedd4l deficiency inhibited VSV-induced IRF3 phosphorylation without affecting the phosphorylation of ERK1/2, p38, JNK1/2, IκBα, and NF-κB p65. Nedd4l deficiency inhibited phosphorylation of TBK1, the upstream molecule that activates IRF3, suggesting that Nedd4l positively regulated RLR signaling by functioning upstream TBK1. In RLR signaling, TBK1 is activated downstream of TRAF3, thus we observed the effect of Nedd4l expression on TRAF3/TBK1 complex formation, which is required for TBK1 activation. In macrophages, Nedd4l deficiency decreased VSV infection-induced TRAF3/TBK1 complex formation (Fig. 3b). In 293T cells, overexpression of Nedd4l increased the association between TRAF3 and TBK1 (Fig. 3c). In contrast, it did not affect the association between TRAF3 and MAVS (Fig. 3d), indicating that Nedd4l regulates VSV-induced interferon production by targeting TRAF3.

TRAF3 not only functions in RLR-mediated type I interferon production, but also functions in TLR-mediated type I interferon

production in macrophages. Similar as in RLR signaling, Nedd4l deficiency inhibited LPS-induced phosphorylation of TBK1 and IRF3 without visibly affecting LPS-induced MAPK and NF-κB phosphorylation (Supplementary Fig. 3a). TRAF3 regulates c-Rel expression in LPS-stimulated macrophages[21]. Nedd4l deficiency decreased c-Rel expression in LPS-stimulated macrophages (Supplementary Fig. 3a), whereas Nedd4l deficiency did not obviously affect c-Rel expression in VSV-infected macrophages (Supplementary Fig. 3b).

**Nedd4l directly interacts with TRAF3**. To illuminate the mechanism by which Nedd4l promotes the interaction between TRAF3 and TBK1, we investigated whether Nedd4l interacts with MAVS, TRAF3, or TBK1. Nedd4l-expressing plasmid was transfected into 293T cells together with plasmids expressing MAVS, TRAF3, or TBK1. Among the three molecules, only TRAF3 was co-immunoprecipitated with Nedd4l (Fig. 4a). Interaction between endogenous TRAF3 and Nedd4l was also detectable in macrophages following virus infection (Fig. 4b). Furthermore, recombinant Nedd4l bound to glutathione S-transferase (GST)-fused TRAF3 in pull-down experiment (Fig. 4c), demonstrating direct

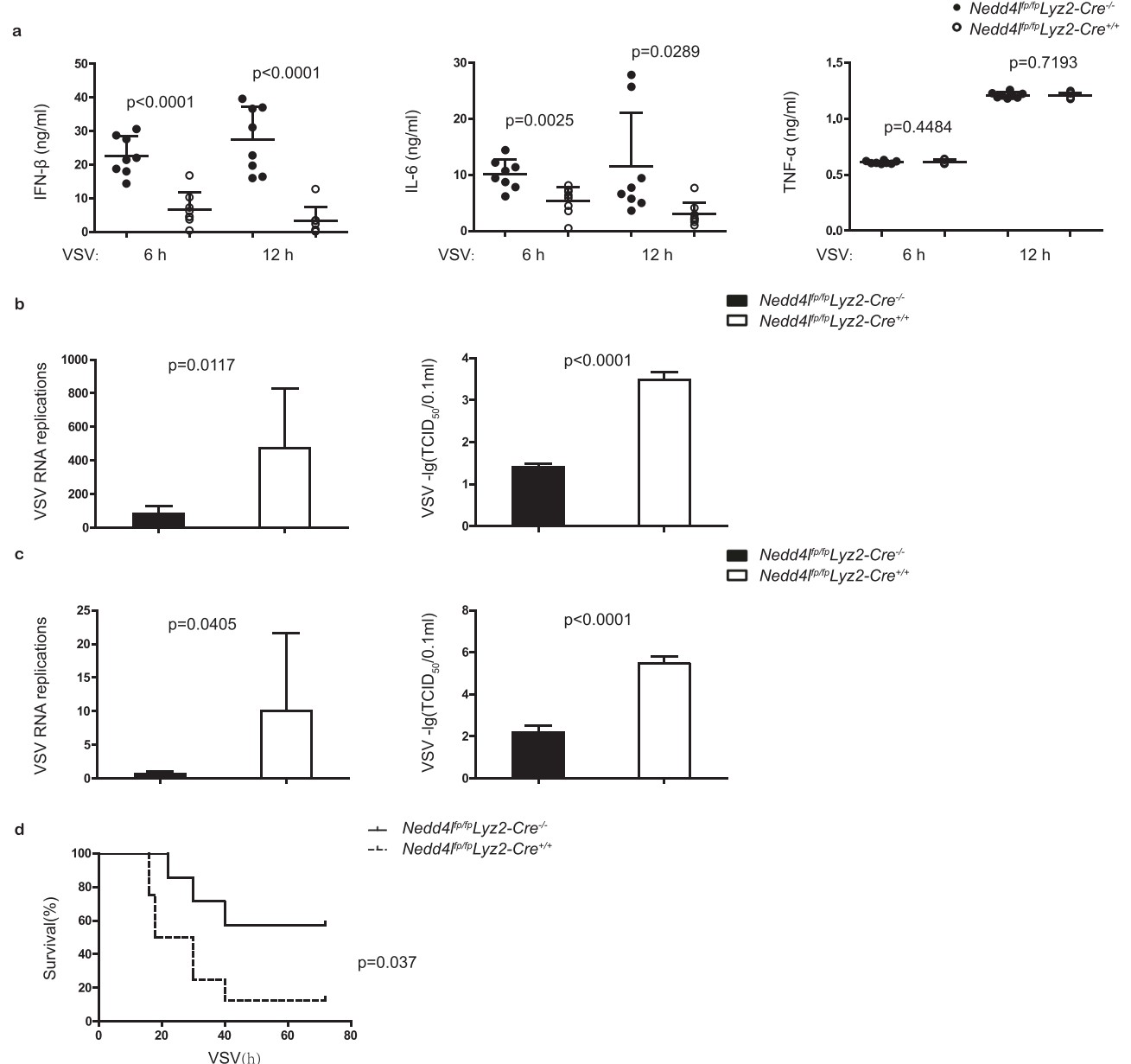

**Fig. 2 Conditional Nedd4l deficiency inhibited antiviral innate immunity in vivo. a** ELISA assay of IFN-β, IL-6, and TNF-α in serum of control *Nedd4l^{fp/fp}Lyz2-Cre^{-/-}* and *Nedd4l^{fp/fp}Lyz2-Cre^{+/+}* mice intraperitoneally infected with VSV ($4 \times 10^5$ PFU/g) for 6 and 12 h ($n = 8$ per group). **b, c** RT-qPCR analysis of VSV mRNA and TCID$_{50}$ in spleen (**b**) and liver (**c**) in control *Nedd4l^{fp/fp}Lyz2-Cre^{-/-}* and *Nedd4l^{fp/fp}Lyz2-Cre^{+/+}* mice intraperitoneally infected with VSV ($4 \times 10^5$ PFU/g) for 24 h (RT-qPCR analysis $n = 8$ and TCID$_{50}$ $n = 6$). **d** The survival rates of control *Nedd4l^{fp/fp}Lyz2-Cre^{-/-}* and *Nedd4l^{fp/fp}Lyz2-Cre^{+/+}* mice were monitored for 3 days after intraperitoneally infected with VSV ($8 \times 10^5$ PFU/g) (*Nedd4l^{fp/fp}Lyz2-Cre^{-/-}* $n = 7$ and *Nedd4l^{fp/fp}Lyz2-Cre^{+/+}* $n = 8$, and *p*-values by two-tailed unpaired Student's *t*-test are indicated). Data are presented as mean ± SD and *p*-values by two-tailed unpaired Student's *t*-test are indicated in (**a–c**) and *p*-value by Gehan-Breslow-Wilcoxon test are indicated in (**d**). Results in (**a–d**) are representative of three independent experiments.

interaction between Nedd4l and TRAF3. Mapping the domain that mediated the binding of Nedd4l to TRAF3, a series of truncated Nedd4l mutants were constructed (Supplementary Fig. 4). Deletion of the C2 or HECT domain in Nedd4l did not affect the interaction between Nedd4l and TRAF3, however, deletion of both C2 and WW domain in Nedd4l disrupted the association between Nedd4l and TRAF3, suggesting that Nedd4l bound TRAF3 through its WW domain (Fig. 4d).

**Nedd4l mediates K29-linked ubiquitination of TRAF3.** Ubiquitination of TRAF3 is required for virus-induced TRAF3/TBK1

complex formation. Since Nedd4l interacts with TRAF3, we supposed that Nedd4l increases TRAF3/TBK1 complex formation by promoting TRAF3 ubiquitination. As shown in Fig. 5a, Nedd4l deficiency decreased VSV infection-induced TRAF3 ubiquitination in macrophages. By using antibodies specific to K48- and K63-linked ubiquitin chains, we found that Nedd4l deficiency decreased both K48- and K63-linked ubiquitination of TRAF3. Nedd4l deficiency also reduced LPS-induced TRAF3 ubiquitination, including K48- and K63-linked ubiquitination in macrophages (Supplementary Fig. 5a). Consistently, Nedd4l overexpression increased TRAF3 ubiquitination, including K48- and K63-linked ubiquitination in VSV-infected 293T cells (Fig. 5b). However, Nedd4l deficiency did

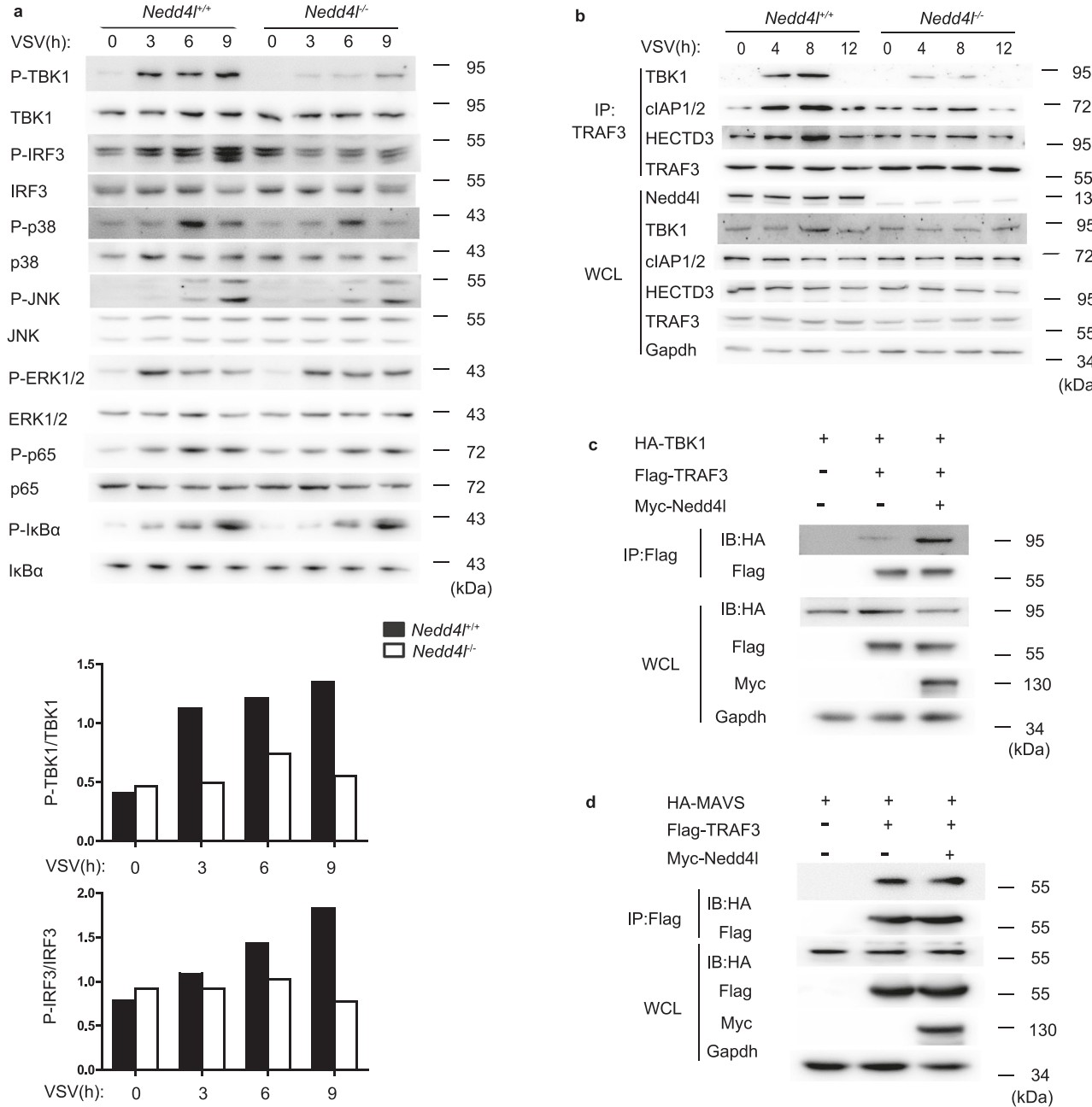

**Fig. 3 Nedd4l promotes TRAF3-dependent signaling. a** Peritoneal macrophages from wild-type (*Nedd4l*[+/+]) and Nedd4l-deficient (*Nedd4l*[−/−]) mice were infected with VSV (MOI = 10) for indicated time. Phosphorylated and total TBK1, IRF3, p38, JNK, ERK1/2, NF-kB p65, and IκBα were detected by western blot. Intensities of p-TBK1, TBK1, p-IRF3, and IRF3 signals in the three independent experiments were quantified by ImageJ and averages of the signals were shown in graphs. **b** Wild-type (*Nedd4l*[+/+]) and Nedd4l-deficient (*Nedd4l*[−/−]) peritoneal macrophages were infected with VSV (MOI = 10) for indicated time, followed by immunoprecipitation (IP) with anti-TRAF3 and immunoblot (IB) analysis with antibodies specific for TBK1, cIAP1/2, HECTD3, TRAF3, Nedd4l, and GAPDH (WCL: whole-cell lysates). **c** IB of Myc-TBK1, Flag-TRAF3, and Myc-Nedd4l co-immunoprecipitated with anti-Flag from lysates of 293T cells transfected with indicated plasmids. **d** IB of HA-MAVS, Flag-TRAF3, and Myc-Nedd4l co-immunoprecipitated with anti-Flag from lysates of 293T cells transfected with indicated plasmids. Results in (**a**–**d**) are representative of three independent experiments.

not reduce LPS-induced TRAF2 ubiquitination (Supplementary Fig. 5b). In vitro ubiquitin ligase activity assay confirmed the ability of Nedd4l to enhance TRAF3 ubiquitination (Fig. 5c). The ability of Nedd4l to increase ubiquitination of TRAF3 was dependent on the ligase activity of Nedd4l, since mutation of C942A or deletion of HECT domain of Nedd4l, both of which disrupted ligase activity of Nedd4l, eliminated Nedd4l-induced increase of TRAF3 ubiquitination (Fig. 5d). On the other hand, deletion of C2 domain increased Nedd4l-mediated ubiquitination, consistently with previous report that C2 domain inhibited Nedd4l ligase activity[22]. A series of HA-

tagged Ub mutants (K11R, K29R, K33R, K48R, K63R), in which one of the seven lysine residues was mutated, were used to identify the lysine residue in ubiquitin that was required for Nedd4l-catalyzed polymeric Ub chain formation. As shown in Fig. 5e, only the mutation of K29 disrupted Nedd4l-mediated TRAF3 ubiquitination. Another series of Ub mutants, in which only one of the seven lysine residues was reserved and the other lysine residues were mutated, were also used to characterize the type of Nedd4l-catalyzed poly-ubiquitin chain. Consistent with the above observation, there was only one mutant, Ub(K29 only), which could be linked to TRAF3 as

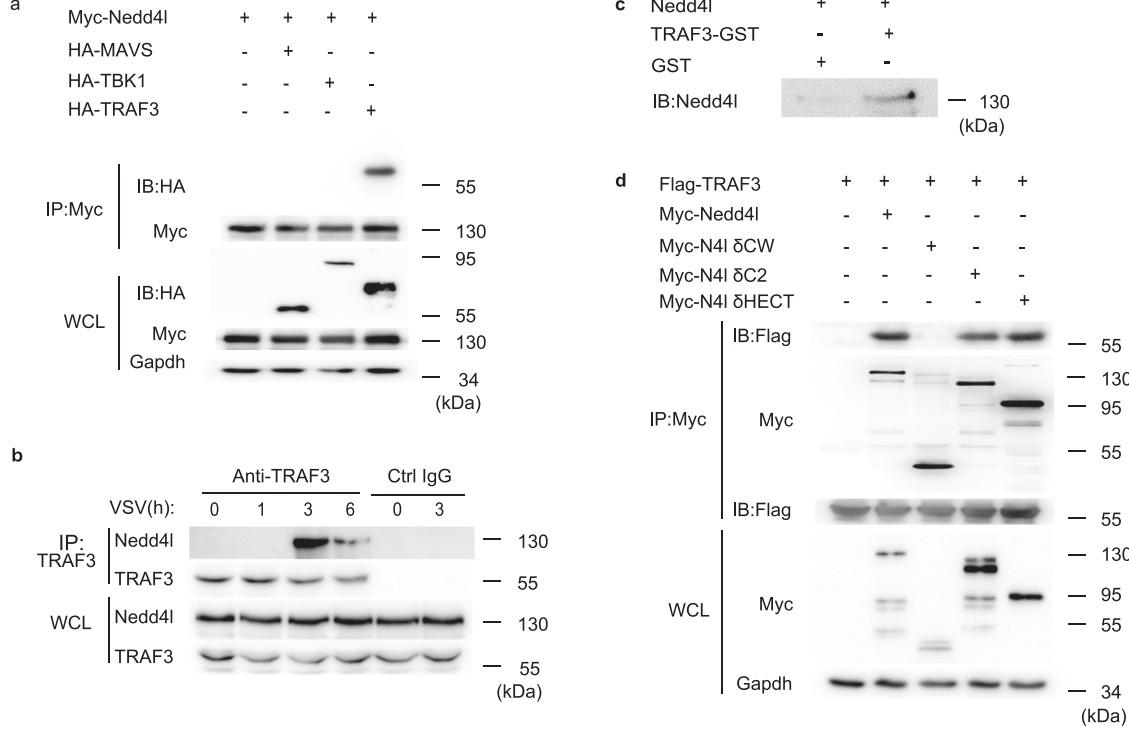

**Fig. 4 Nedd4l interacts with TRAF3. a** IB of HA-MAVS, HA-TBK1, HA-TRAF3, and Myc-Nedd4l co-immunoprecipitated with anti-Myc from lysates of 293T cells transfected with indicated plasmids with HA-tag, Myc-tag, or GAPDH antibodies. **b** IB of TRAF3 and Nedd4l co-immunoprecipitated with anti-TRAF3 from lysates of C57BL/6 mice peritoneal macrophages infected with VSV (MOI = 10) for indicated time. **c** In vitro GST-pull-down assay of Nedd4l with GST or GST-fused TRAF3. **d** IB of TRAF3 and Nedd4l co-immunoprecipitated with Myc-tag-specific antibody from lysates of 293T cells co-transfected with plasmids expressing Flag-TRAF3 and Myc-tagged wild-type Nedd4l or mutant Nedd4l with CW (ΔCW),C2 (ΔC2), or HECT (ΔHECT) domain deleted. Results in (**a–d**) are representative of three independent experiments.

efficiently as wild-type ubiquitin (Fig. 5f), demonstrating that Nedd4l directly catalyzed K29-linked TRAF3 ubiquitination rather than directly catalyzing K48- or K63-linked TRAF3 ubiquitination. We hypothesize that Nedd4l-catalyzed K29-linked ubiquitination might facilitate K48- and K63-linked self-ubiquitination of TRAF3 or ubiquitination of TRAF3 mediated by other E3 ligases.

**Nedd4l ubiquitinates cysteine residues in TRAF3.** We tried to identify Nedd4l-catalyzed ubiquitination sites in TRAF3. TRAF3 plasmid was transfected into 293T cells with or without Nedd4l plasmid, and then TRAF3 was immunoprecipitated for liquid chromatography-tandem mass spectrometry (LC-MS/MS) analysis. The LC-MS/MS data suggested that Nedd4l overexpression increased ubiquitination at K273 and K315 lysine residues (Supplementary Fig. 6a, b). We mutated these candidate ubiquitination sites, transfected the mutants into 293T cells together with Nedd4l and mutant HA-tagged Ub (K29 only). K50 residue was also mutated as a negative control. As shown in Fig. 6a, mutation of K50, K273, and K315 did not affect K29-linked ubiquitination of TRAF3. Interestingly, LC-MS/MS analysis showed that ubiquitination at two cysteine residues, C56 and C124, also increased in Nedd4l overexpressing cells (Supplementary Fig. 6c, d). TRAF3 contains six zinc fingers, two of them (RZ1 and RZ2) are located in the RING (really interesting gene) domain, the other four (Z1, Z2, Z3, and Z4) are located between the RING domain and C-terminal TRAF domain (Supplementary Fig. 6e). C56, together with C53, C73, and C76, forms a zinc finger in the N-terminal RING domain (RZ1). Meanwhile, C124, together with C117, H136, and C141, forms a zinc finger (Z1) between the RING domain and C-terminal TRAF domain (Supplementary Fig. 6e).

We constructed mutant TRAF3(C56R) and TRAF3(C124R) plasmids. Surprisingly, mutation of C56 and C124 dramatically decreased Nedd4l-catalyzed K29-linked ubiquitination of TRAF3 (Fig. 6a), suggesting that the C56 and C124 residues were the ubiquitination sites to which Nedd4l-catalyzed k29-linked ubiquitin chains were attached. However, when the TRAF3 mutants were transfected with Nedd4l and wild-type HA-tagged ubiquitin (HA-Ub), mutation of K273 and K315 slightly decreased TRAF3 ubiquitination, but mutation of C56 and C124 remarkably increased TRAF3 ubiquitination, including K48- and K63-linked ubiquitination (Fig. 6b). To illuminate the mechanism underlying the phenomenon, we observed the effects of C56 and C124 mutation on TRAF3 ubiquitination without Nedd4l overexpression. Interestingly, mutation of C56 or C124 increased K48- and K63-linked ubiquitination of TRAF3 even in the absence of Nedd4l overexpression (Fig. 6c). Since C56 and C124 are residues that constitute RZ1 and Z1 zinc fingers, we suppose that C56 and C124 mutation might increase K48- and K63-linked ubiquitination of TRAF3 via affecting the formation of the zinc fingers. To further investigate the role of RZ1 and Z1 zinc fingers in the regulation of TRAF3 ubiquitination, we mutated the other residues (C53R, C73R, C76R, C117R, H136R, and C141R) which formed RZ1 and Z1 zinc fingers together with C56 and C124. Mutating each of the residues significantly increased TRAF3 ubiquitination (Fig. 6d), suggesting that modulation or disruption of RZ1 and Z1 zinc fingers increases TRAF3 ubiquitination.

TRAF3 is an E3 ligase and proposed to be capable of self-ubiquitination. Mutation of C68A/H70A inactivates the ligase activity of TRAF3[12]. To examine whether self-ubiquitination was responsible for the increased ubiquitination in TRAF3(C56R) and TRAF3(C124R), we introduced C68A/H70A mutation to

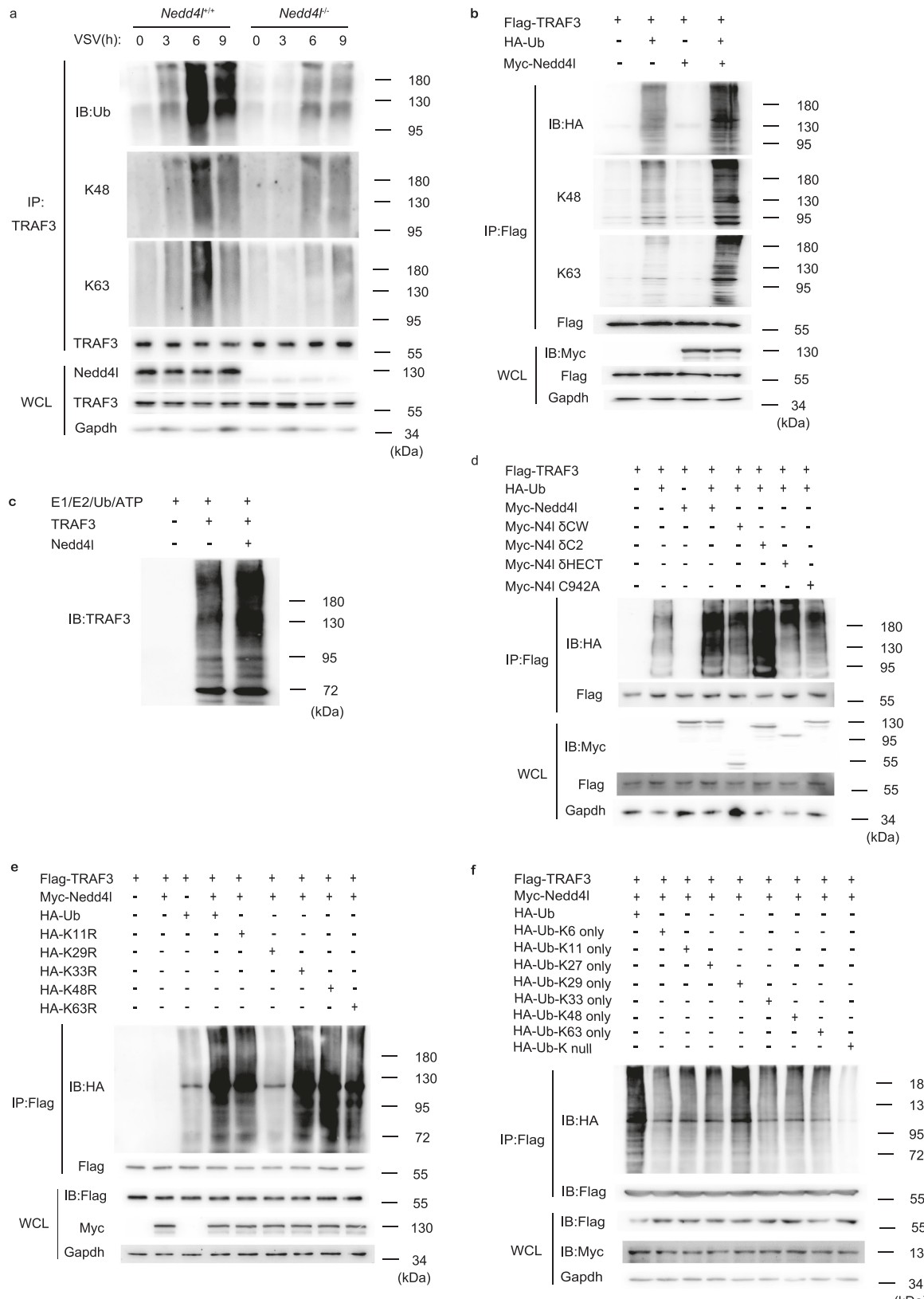

construct ligase activity dead TRAF3(C56R) and TRAF3(C124R). As shown in Fig. 6c, catalytic dead mutant TRAF3(C56R/C68A/H70A) and TRAF3(C124R/C68A/H70A) were similarly ubiquitinated as TRAF3(C56R) and TRAF3(C124R), suggesting that self-ubiquitination was not the major cause for the increased TRAF3(C56R) and TRAF3(C124R) ubiquitination.

**Nedd4l promotes TRAF3 to interact with cIAP1/2 and HECTD3**. TRAF3 is ubiquitinated by other E3 ligases such as cIAP1/2 and HECTD3[12,18]. We examined whether Nedd4l affected the interaction between TRAF3 and these E3 ligases. As shown in Fig. 3b, Nedd4l deficiency decreased VSV infection-induced TRAF3-cIAP1/2 and TRAF3-HECTD3 complex

**Fig. 5 Nedd4l-catalyzed K29-linked ubiquitination of TRAF3. a** IB of total ubiquitination, K48- and K63-linked ubiquitination of TRAF3 in VSV-infected wild-type (Nedd4l$^{+/+}$) and Nedd4l-deficient (Nedd4l$^{-/-}$) peritoneal macrophages. **b** Plasmids expressing Flag-TRAF3, HA-Ub, and Myc-Nedd4l were co-transfected in 293T cells. After 24 h, cells were infected with VSV (MOI = 10) for 12 h. IB analysis of total ubiquitination, K48- and K63-linked ubiquitination of Flag-tagged TRAF3 with indicated antibodies. **c** IB analysis of TRAF3 incubated with Nedd4l recombinant proteins and ubiquitination reaction components in vitro. **d** IB analysis of total ubiquitination of Flag-tagged TRAF3 in 293T cells co-transfected with plasmids expressing Flag-TRAF3, HA-Ub, and Myc-tagged wild-type Nedd4l or mutant Nedd4l with indicated antibodies. **e** IB analysis of ubiquitination of Flag-TRAF3 in 293T cells co-transfected with plasmids expressing Flag-TRAF3, Myc-Nedd4l, and HA-tagged wild-type Ub or HA-tagged mutant Ub (K11R, K29R, K33R, K48R, and K63R) with indicated antibodies. **f** IB analysis of ubiquitination of Flag-TRAF3 in 293T cells co-transfected with plasmids expressing Flag-TRAF3, Myc-Nedd4l, and HA-tagged wild-type Ub or HA-tagged mutant Ub (K6 only, K11 only, K27 only, K29 only, K33 only, K48 only, K63 only, Ub-K null) with indicated antibodies. Results in (**a**–**f**) are representative of three independent experiments.

formation in macrophages. Consistently, in 293T cells, over-expression of Nedd4l promoted TRAF3 to associate with cIAP1/2 and HECTD3 (Fig. 7a, b). Similar to the results in ubiquitination assay, mutation of C56 also increased the interaction of TRAF3 with cIAP1/2 and HECTD3 (Fig. 7c, d). The effects of C56 and C124 mutation on TRAF3-mediated signaling were examined. When overexpressed in HEK293T cells, mutant TRAF3(C56R) and TRAF3(C124R) more efficiently recruited TBK1 to form TRAF3/TBK1 complex compared with wild-type TRAF3 (Supplementary Fig. 7a). We co-transfected wild-type TRAF3, mutant TRAF3(C56R), or TRAF3(C124R) together with NF-κB or IRF3 luciferase reporter gene into HEK293 cells. Compared with wild-type TRAF3, mutant TRAF3(C56R) and TRAF3(C124R) increased NF-κB and IRF3 luciferase reporter gene expression in a dose-dependent manner (Supplementary Fig. 7b, c). Consistently, mutation of C56 and C124 increased VSV-induced IFN-β, IL-6, and TNF-α mRNA expression (Fig. 7e). C61 is near to C56. However, mutation of C61 did not affect VSV-induced IFN-β, IL-6, and TNF-α mRNA expression (Fig. 7e). TRAF3, TRAF3(C56R), and TRAF3(C124R) plasmids were also transfected into TRAF3-deficient HEK293T cells. The doses of the transfected plasmids were optimized so that expression level of TRAF3, TRAF3(C56R), and TRAF3(C124R) in the cells was comparable with that of endogenous TRAF3 in wild-type parent HEK293T cells (Supplementary Fig. 7d). In these cells, mutation of C56 and C124 also increased VSV-induced IFN-β mRNA expression (Supplementary Fig. 7e), providing further evidence that C56 and C124 residues are important in regulating virus-induced innate immunity.

## Discussion

In the present study, we demonstrate that Nedd4l promotes antiviral innate immunity by showing that Nedd4l deficiency impairs VSV-induced production of proinflammatory cytokine and type I interferon, and Nedd4l-deficient mice are more susceptible to VSV infection. Nedd4l deficiency inhibits VSV-induced TRAF3 ubiquitination, TRAF3/TBK1 complex formation, and IRF3 phosphorylation. Interestingly, Nedd4l promotes type I interferon signals by catalyzing K29-linked ubiquitination of cysteine residue in TRAF3.

TRAF3 selectively regulates expression of type I interferons and proinflammatory cytokines in PRR signaling via modification by different modes of ubiquitination[12]. While K48-linked ubiquitination of TRAF3 promotes proinflammatory cytokine production, K63-linked ubiquitination of TRAF3 increases expression of type I interferon. The present study reveals another mode of ubiquitination, that is K29-linked ubiquitination, which regulates the activity of TRAF3 to promote innate immunity.

Several lines of evidences support the conclusion that Nedd4l directly catalyzes K29-linked ubiquitination of TRAF3 and subsequently indirectly promotes K48- or K63-linked ubiquitination of TRAF3. When ubiquitin with one lysine residue mutation is used to identify the lysine residue linking the polyubiquitin chain

to TRAF3, except Ub(K29R), none of the other mutant ubiquitins inhibits Nedd4l-mediated TRAF3 ubiquitination. Consistently, the mutant Ub(K29 only), but not the other mutant ubiquitins, with only one lysine residue reserved, links to TRAF3 as efficiently as wild-type ubiquitin. Nedd4l deficiency decreases virus-induced TRAF3 ubiquitination, including K48- and K63-linked ubiquitination in macrophages (Fig. 5a), and Nedd4l over-expression increases K48- and K63-linked TRAF3 ubiquitination in virus-infected HEK293T cells (Fig. 5b). LC-MS/MS analysis and ubiquitination assay identify C56 and C124 as the ubiquitination sites targeted by Nedd4l. However, C56R and C124R mutation prevent Nedd4l-mediated K29-linked TRAF3 ubiquitination, but increase K48- and K63-linked TRAF3 ubiquitination even in the absence of Nedd4l overexpression. These results are not contradictory, but raise a model that Nedd4l-mediated K29-linked ubiquitination indirectly promotes PRR-triggered K48- and K63-linked TRAF3 ubiquitination. In this model, PRR signal induces Nedd4l to interact with TRAF3 and catalyze K29-linked ubiquitination of C56 and C124 in TRAF3, ubiquitination of the cysteines subsequently increases K48- and K63-linked TRAF3 ubiquitination catalyzed by other E3 ligases. Supporting the model, Nedd4l enhances the interaction between TRAF3 and other E3 ligases, such as cIAP1/2 and HECTD3. Self-ubiquitination activity of TRAF3 is not required for the increased K48- and K63-linked ubiquitination, since the increased ubiquitination is not affected by C68A/H70A mutation, which inactivates the TRAF3 ligase activity.

Interestingly, both C56 and C124 residues, which are ubiquitinated by Nedd4l, contribute to the formation of two zinc fingers, RZ1 and Z1, in TRAF3. It is possible that Nedd4l-mediated K29-linked ubiquitination of C56 and C124 affect the conformation of the zinc fingers, and changes of the zinc fingers induced by Nedd4l-mediated ubiquitination promote TRAF3 to interact with cIAP1/2 and HECTD3, resulting in increased K48- and K63-linked TRAF3 ubiquitination. Consistent with the hypothesis, C56R and C124R mutations, which not only dispossess the sites of K29-linked ubiquitination, but also disrupt the formation of the zinc fingers, enhance the interaction between TRAF3, cIAP1/2 and HECTD3, and increase K48- and K63-linked TRAF3 ubiquitination. In fact, mutating each of the eight residues that contribute to the formation of RZ1 and Z1 zinc fingers can increase K48- and K63-linked ubiquitination of TRAF3 (Fig. 6d). In contrast, mutation of C61, which is nearby C56R but is not required for the formation of the zinc fingers, does not affect VSV-induced type I interferon signaling (Fig. 7e). These results demonstrate that the two zinc fingers play important roles in regulating TRAF3 ubiquitination. Zinc fingers widely exist in proteins. It is worthy to investigate whether ubiquitination of cysteine also happens to other proteins containing zinc fingers and regulates functions of the proteins.

Ubiquitination usually occurs on lysine residues of substrate proteins. However, several studies reported ubiquitination on cysteine. Ubiquitination on cysteine was initially found in MHC-I

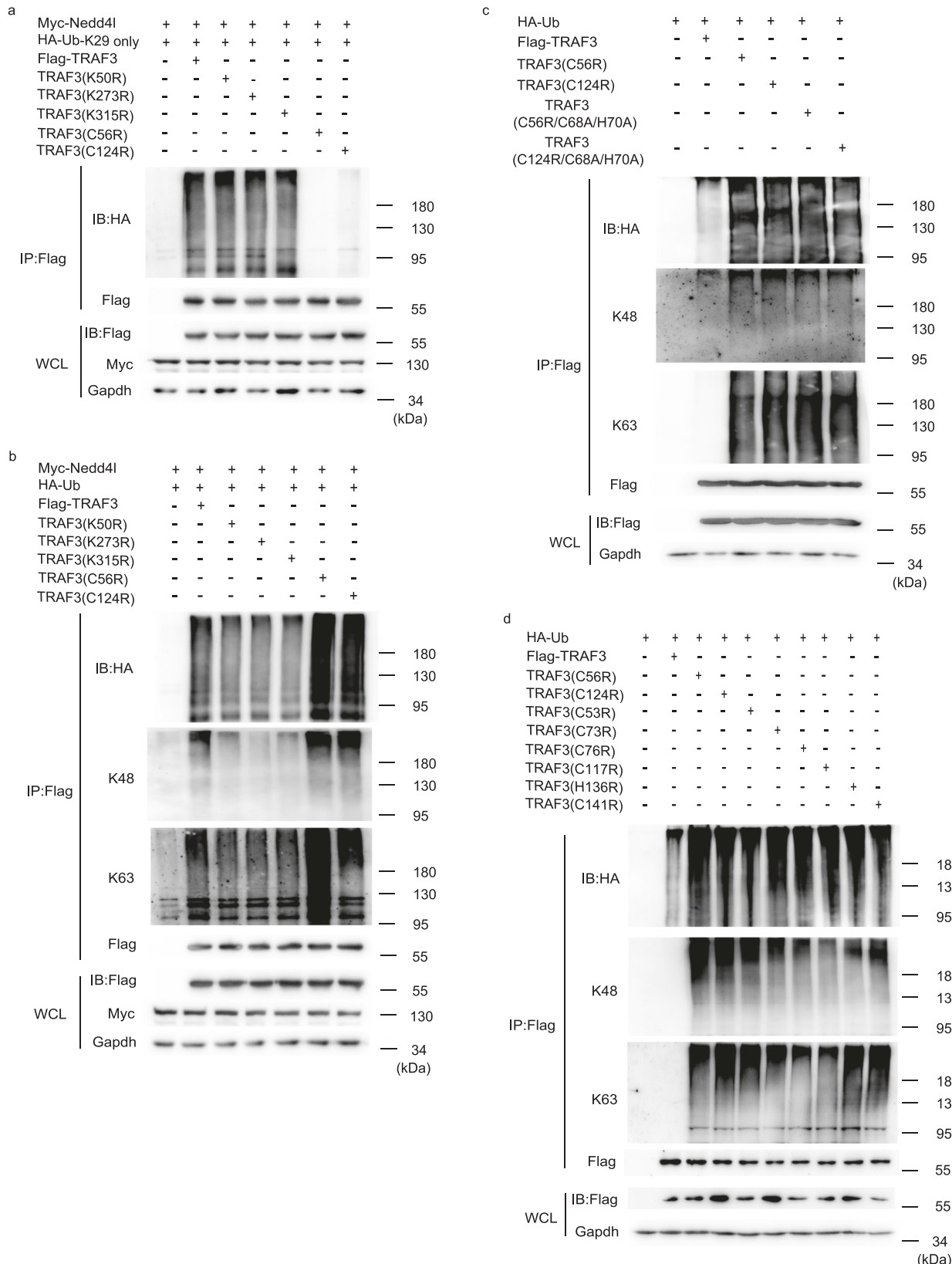

molecules, which was mediated by Kaposi's sarcoma-associated herpes virus-encoded E3 ubiquitin ligase MIR1 and down-regulated MHC I molecules[3]. Suv39h1, a H3K9me3-specific histone methyltransferase, is ubiquitinated on conserved cysteines, and mutation of the cysteines impairs the function of Suv39h1 to regulate activation of NF-κB pathway[23]. Acetyl-CoA acetyltransferase 2, a cellular enzyme which converts cholesterol and fatty acid to cholesteryl esters, is ubiquitinated on Cys277 by Lys48-linked polyubiquitylation for degradation when the lipid level is low, and thus regulates lipid homeostasis[24]. Despite these findings, the physiological significance of cysteine ubiquitination is seriously underestimated. Although lysine ubiquitination in PRR signaling has been

**Fig. 6 Nedd4l ubiquitinates cysteine residues in TRAF3. a** IB analysis of ubiquitination of Flag-tagged TRAF3 in 293T cells co-transfected with plasmids expressing Myc-Nedd4l, HA-tagged mutant Ub (K29 only), and Flag-tagged wild-type TRAF3 or mutant TRAF3 (K50R, K273R, K315R, C56R, or C124R). **b** IB analysis of total ubiquitination, K48- and K63-linked ubiquitination of Flag-tagged TRAF3 in 293T cells co-transfected with plasmids expressing Myc-Nedd4l, HA-tagged wild-type Ub, and Flag-tagged wild-type TRAF3 or mutant TRAF3 (K50R, K273R, K315R, C56R, or C124R). **c** IB analysis of total ubiquitination, K48- and K63-linked ubiquitination of Flag-tagged TRAF3 in 293T cells co-transfected with plasmids expressing HA-tagged wild-type Ub and Flag-tagged TRAF3, TRAF3 (C56R), TRAF3 (C124R), TRAF3(C56R/C68A/H70A), or TRAF3(C124R/C68A/H70A). **d** IB analysis of total ubiquitination, K48- and K63-linked ubiquitination of Flag-tagged TRAF3 in 293T cells co-transfected with plasmids expressing HA-tagged wild-type Ub and Flag-tagged wild-type TRAF3 or mutant TRAF3 (C56R, C124R, C53R, C73R, C76R, C117R, H136R, C141R). Results in (**a–d**) are representative of three independent experiments.

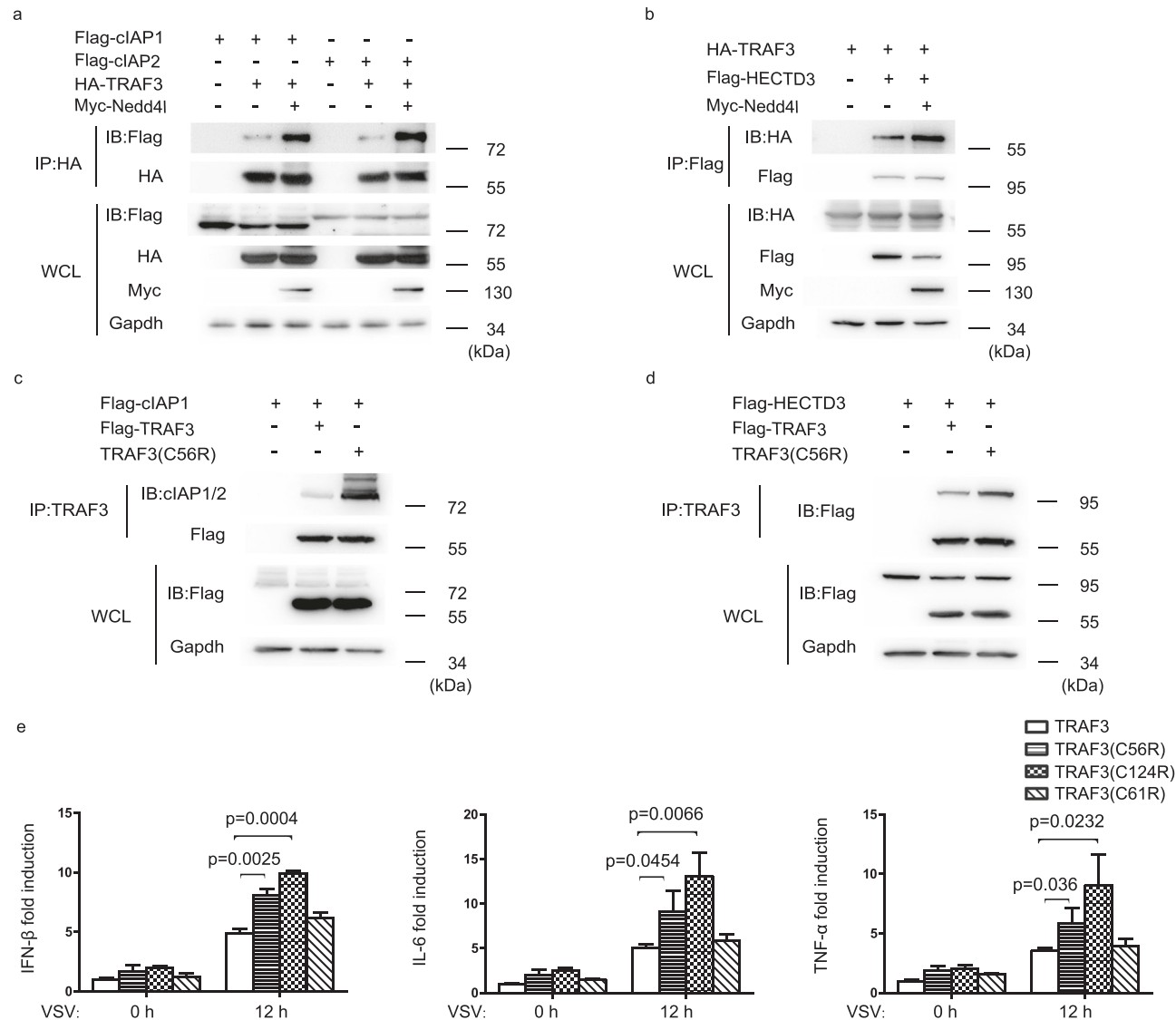

**Fig. 7 Nedd4l promotes TRAF3 to associate with cIAP1/2 and HECTD3. a** IB of Flag-cIAP1, Flag-cIAP2, Myc-Nedd4l, and HA-TRAF3 co-immunoprecipitated with anti-HA from lysates of 293T cells transfected with indicated plasmids. **b** IB of Flag-HECTD3, Myc-Nedd4l and HA-TRAF3 co-immunoprecipitated with anti-HA from lysates of 293T cells transfected with indicated plasmids. **c** IB of Flag-cIAP1, Flag-TRAF3, and Flag-TRAF3(C56R) co-immunoprecipitated with anti-TRAF3 from lysates of 293T cells transfected with indicated plasmids. **d** IB of Flag-HECTD3, Flag-TRAF3 and Flag-TRAF3 (C56R) co-immunoprecipitated with anti-TRAF3 from lysates of 293T cells transfected with indicated plasmids. **e** RT-qPCR analysis of IFN-β, IL-6, and TNF-α mRNA expression in 293 cells transfected with plasmids expressing TRAF3 or mutant TRAF3 and then infected with VSV (MOI = 10) for 12 h. Data are presented as mean ± SD (*n* = 3 per group) and *p*-values by two-tailed unpaired Student's *t*-test are indicated in e. Results in (**a–e**) are representative of three independent experiments.

intensively studied and demonstrated to be important in the regulation of innate immunity[8,9], cysteine ubiquitination in PRR signaling has not yet been found. Our results demonstrate that cysteine ubiquitination plays an important role in antivirus innate immunity. The mechanism and significance of cysteine ubiquitination in innate immunity merits further investigation.

Nedd4l deficiency also results in decreased proinflammatory cytokine production in macrophages. Although Nedd4l deficiency

inhibits virus- and LPS-induced TBK1 and IRF3 phosphorylation, Nedd4l deficiency does not affect phosphorylation of NF-κB p65, ERK1/2, p38, JNK, and IκBα. TRAF3 inhibits LPS-induced proinflammatory cytokine production by reducing IRF5 and c-Rel expression[21,25–28]. Nedd4l deficiency inhibits expression of c-Rel in LPS-stimulated macrophages. However, c-Rel expression is not affected by Nedd4l deficiency in VSV-infected macrophages (Supplementary Fig. 3b). c-Rel deficiency selectively decreases IL-12p40 expression, but does not impair TNF-α and IL-6 production[26,27]. It seems unlikely that Nedd4l increases LPS- and VSV-induced proinflammatory cytokine production via regulating c-Rel and IRF5 expression. The exact mechanism by which Nedd4l regulates proinflammatory cytokine production needs illumination in further investigation. It was reported that c-Rel negatively regulate TLR3-mediated IFN-β production[29]. However, a recent study showed that MEF cells and BMDMs expressing mutant truncated c-Rel expressed comparable levels of IFN-β with wild-type control MEF cells and BMDMs following HSV-1 infection[30]. The role of c-Rel in IFN-β production in innate immunity needs more investigation to confirm.

In addition to poly(I:C)-induced IFN-β production, Nedd4l deficiency also impairs poly(dA:dT)-induced IFN-β production. It remains unclear how Nedd4l increases poly(dA:dT)-induced IFN-β production. TRAF3 inhibits STING-dependent IFN-β production by degrading NIK, which interacts with STING to enhance IFN induction[31]. The present study demonstrates that Nedd4l positively regulates poly(I:C)-induced IFN-β production by promoting TRAF3 ubiquitination. Ubiquitination of TRAF3 by Nedd4l promotes interaction of TRAF3 with proteins such as cIAP1/2, HECTD3, and TBK1. These data do not mean that ubiquitination of TRAF3 by Nedd4l will promote TRAF3-mediated NIK degradation. In this case, Nedd4l may also positively regulate poly(dA:dT)-induced IFN-β production by promoting TBK1 activation. It is worth investigating whether and how ubiquitination of TRAF3 by Nedd4l regulates NIK degradation. Besides, there is also a possibility that Nedd4l regulates poly(dA:dT)-induced IFN-β production by targeting other molecules rather than TRAF3 in DNA-sensing pathway. The exact mechanism by which Nedd4l increases poly(dA:dT)-induced IFN-β production needs further investigation.

In conclusion, the present study demonstrates that Nedd4l promotes VSV- and LPS-induced type I interferon production by catalyzing K29-linked ubiquitination of cysteine residues in TRAF3, not only identifying Nedd4l as a critical regulator of antiviral innate immunity, but also bringing to light a much more complex regulatory landscape of TRAF3 ubiquitination than the previously anticipated one. Importantly, this study reveals the significance of cysteine ubiquitination in innate immunity. Investigating whether ubiquitination of cysteines is also present in other molecules in PRR signaling will bring insight into the mechanism by which innate immunity is activated and regulated.

## Methods

**Cell culture and reagents.** HEK293 cells (ATCC: CRL-1573) and HEK293T cells (ATCC: RCB-2202) were provided by Shanghai Zhong Qiao Xin Zhou Biotechnology Corp., Ltd., and originally purchased from American Type Culture Collection (ATCC), and maintained in DMEM (Hyclone) with 10% (vol/vol) fetal bovine serum (FBS) at 37 °C with 5% $CO_2$. Antibodies specific for phosphorylated TBK1 (Ser172,5483), phosphorylated IRF3 (Ser396,4947), phosphorylated ERK1/2 (Thr202/Tyr204,9101), phosphorylated p38 (Thr180/Tyr182,9211), phosphorylated NF-κB p65 (Ser536,3033), phosphorylated JNK (Thr183/Tyr185,9251), phosphorylated IκBα (Ser32,2859), TBK1 (3504), IRF3 (4302), JNK (9252), NF-κB p65 (8242), IκBα (4814), Nedd4l (4013), TRAF3 (4729), c-Rel (12707), Ubiquitin (3936), K63-linkage-specific polyubiquitin (5621), K48-linkage-specific polyubiquitin (4289), and Myc (71D10) were from Cell Signaling Technology. Antibodies specific for β-actin (sc-1616), normal rabbit IgG (sc-2027), p38 (sc-535), TRAF2 (sc-877), cIAP1/2 (sc-12410), and ERK1/2 (sc-93) were from Santa Cruz Biotechnology. These antibodies were applied at the dilution 1:1000. Flag Tag (F1804) and HA Tag (26183) were from Sigma and applied at the dilution 1:10,000. Anti-GAPDH (6004) antibody and HECTD3 (11487-1-AP) were from Proteintech and applied at the dilution 1:20,000 and 1:1000, respectively. CD4-PE (100408), CD19-APC (302211), and CD11b-APC (101212)

were from BioLegend and applied at the dilution 1:100. CD8-FITC (11-0081-82), Ly-6G(Gr-1)-PE (12-5931-82), F4/80-APC (17-4801-82), and CD11b-FITC (11-0112-81) were from eBioscience and applied at the dilution 1:100. CD3-FITC (561827) was from BD and applied at the dilution 1:100. CD4 MicroBeads (130-049-201) and CD19 MicroBeads (130-052-201) were from MiltenyiBiotec. LPS (437628) was from Calbiochem. LTA (L2515) and Protein G Sepharose (P3296) were from Sigma. GST-TRAF3 (3384H) was from Creative Biomart. Recombinant Nedd4l (TP327866) and recombinant TRAF3 (TP318682) were from Origene. E1 (304), E2 (616), and Ub (U-100H) were from Boston Biochem. Poly(I:C) and poly(dA:dT) were from Calbiochem. Protein marker (26617) was from Thermo Fisher Scientific.

**Preparation of peritoneal macrophages.** C57BL/6 mice were obtained from Joint Ventures Sipper BK Experimental Animal Company. Nedd4l-deficient mice were purchased from Jaxmice. Nedd4l conditional knockout mice were purchased from Guangzhou Cyagen Biosciences. Lyz2-Cre mice were purchased from Shanghai Biomodel Organism Science and Technology Development Limited Company. Nedd4l-deficient mice and littermate control mice were used in experiments. All studies were conducted in accordance with the National Institute of Health Guide for the Care and Use of Laboratory Animals with the approval of the Scientific Investigation Board of Second Military Medical University, Shanghai. Female C57BL/6 mice (6–8 week old) were injected with 2 ml 3% sodium thioglycolate (Sigma) solution intraperitoneally. Four days later cells in the abdominal cavity were collected by lavaging with 15 ml DMEM without FBS and centrifugation. Cells were seeded and maintained in DMEM with 10% FBS. The adhered peritoneal macrophages were used for further experiments 24 h later[32].

**Quantitative reverse transcription-PCR.** Total RNA of cells was isolated with RNA extraction kit (Fastagen) and RNA of spleen and liver from mice was isolated with Trizol (Invitrogen). cDNA was synthesized by using reverse transcriptase M-MLV (Takara) according to the manufacturer's instructions. Real-time quantitative PCR was performed on ABI7500 (Applied Biosystems) with RT-qPCR kit (Promega). Data of each sample is normalized to actin mRNA expression. The relative expression levels of the genes in control cells are presented as 1 fold. The mRNA expression levels of the assayed genes are normalized to mRNA expression of actin in each sample and presented as fold induction after division with the expression levels of the genes in control cells. The primers used are as follows: β-actin-F 5′-AGTGTGACGTTGACATCCGT-3′; β-actin-B 5′-GCAGCTCAGTAA-CAGTCCGC-3′; IFN-β-F 5′-ATGAGTGGTGGTTGCAGGC-3′; IFN-β-B 5′-TGACCTTTCAAATGCAGTAGATTCA-3′; TNF-α-F 5′-AAGCCTGTAGCC CACGTCGTA −3′; TNF-α-B 5′-GGCACCACTAGTTGGTTGTCTTTG −3′; IL-6-F 5′-TAGTCCTTCCTACCCCAATTTCC-3′; IL-6-B 5′-TTGGTCCTTAGC-CACTCCTTC-3′; VSV-F 5′-ACGGCGTACTTCCAGATGG-3′; VSV-B 5′-CTCGGTTCAAGATCCAGGT-3′.

**Detection of cytokine production.** The concentration of TNF-α (eBioscience), IL-6 (eBioscience), and IFN-β (Bioligend) in culture supernatants and serums were measured with ELISA kit according to the manufacturer's instructions.

**RNA transfection.** Small RNA was transfected by using INTERFERin (Polyplus) following the manufacturer's instructions. Control non-target siRNA and Nedd4l on-target siRNA were from Dharmacon.

**DNA transfection.** Transfections were performed by using the jetPEI transfection reagent (Polyplus) following the manufacturer's instructions. HEK293T cells were seeded 12 h before transfection, and the cells were harvested 24 h after transfection.

**Plasmid constructs.** The plasmid expressing Nedd4l was obtained from Origene. TRAF3 and TBK1 were amplified from mRNA of Hela cells and RAW264.7, respectively, by reverse transcription-PCR and the cDNA expressing full-length TRAF3 or TBK1 were cloned into pcDNA3.1-Flag, pCMV-Myc, or pCMV-HA expression vector (Clontech) to construct Flag-, Myc-, or HA-tagged TRAF3 and TBK1-expressing plasmids[33,34]. Mutant Nedd4l and TRAF3 plasmids were constructed using Takara MutanBEST kit (R401) according to the manufacturer's instructions. All of the constructs were confirmed by DNA sequencing. A series of HA-tagged mutant Ub plasmids were a gift from Mingjin Yang (Institute of Immunology, Second Military Medical University) and Ronggui Hu (State Key Laboratory of Molecular Biology, Shanghai Science Research Center, CAS Center for Excellence in Molecular Cell Science, Shanghai Institute of Biochemistry and Cell Biology, Chinese Academy of Sciences, University of Chinese Academy of Sciences). cIAP1- and cIAP2-expressing plasmids were a gift from Hongbo Hu (State Key Laboratory of Biotherapy, Sichuan University). HECTD3-expressing plasmid was a gift from Xiaopeng Qi (Key Laboratory of Animal Models and Human Disease Mechanisms of the Chinese Academy of Sciences and Yunnan Province).

**Immunoblot and Immunoprecipitation analysis.** Cells were lysed with M-PER Protein Extraction Reagent (Pierce) supplemented with protease inhibitor cocktail and protein concentrations of the extracts were measured with BCA assay kit (Pierce). Equal amounts of proteins were loaded for SDS-PAGE, transferred onto

nitrocellulose membranes, and then blotted with indicated antibodies. The signal intensity was determined using the Tanon 5200s Chemiluminescent Imaging System (Tanon). Images have been cropped for presentation. The data were analyzed using ImageJ software.

**Detection of protein ubiquitination.** Cells were treated with proteasome inhibitors MG132 (Selleck, 30 μM) for 6 h and then lysed. The cell lysates were boiled for 10 min after adding 1% SDS. The cell lysates were diluted to 0.1% SDS with lysis buffer. Protein concentrations of the extracts were measured, and equal amounts of extracts were used for immunoprecipitation of target protein.

**VSV infection model.** Six to eight week old mice were challenged with VSV intraperitoneally (nonlethal dosage: $4 \times 10^5$ PFU/g; lethal dosage: $8 \times 10^5$ PFU/g). Survival of mice were monitored for 3 days. $TCID_{50}$ (50% tissue culture infective dose) of VSV was detected with Karber Method.

**Nanospray liquid chromatography-tandem mass spectrometry.** Nedd4l was immunoprecipitated and separated by SDS-PAGE, and then analyzed by nano-ultra-performance liquid chromatography-electrospray ionization tandem mass spectrometry (Beijing Protein Institute).

**Assay of luciferase reporter gene expression.** HEK293 cells were co-transfected with NF-κB or IRF3 activity luciferase reporter plasmid, pRL-TK-Renilla-luciferase plasmid, and indicated amount of TRAF3(C56R)-, TRAF3(C124R)-, or MAVS-expressing plasmids. After 24 h, luciferase activity in the cell lysates were measured with a Dual-Luciferase Reporter Assay system according to the manufacturer's instructions (Promega). The luciferase activity was normalized for transfection efficiency by division with Renilla luciferase activity[35].

**Flow cytometry and cells isolation.** For flow cytometric analysis of macrophages, granulocytes, B cells and T cells, and single-cell suspensions prepared from spleen, bone marrow, and peripheral blood were stained using a subset of antibodies. The preliminary FSC-A/SSC-A gates were used for the starting cell population and FSC-H/FSC-W gates were used for removing adhesion. Macrophages were stained with F4/80-APC and CD11b-FITC. Granulocytes were stained with Gr-1-PE and CD11b-APC. T cells were stained with CD3-FITC, CD4-PE, and CD8-FITC. B cells were stained with CD19-APC (Supplementary Fig. 2d). Cells were analyzed on a Attune NxT (Invitrogen). The data was analyzed using FlowJo software. For T cell and B cell isolation, single-cell suspensions prepared from spleen and bone marrow were labeled with CD4 microbeads and CD19 microbeads separately within a MACS Column according to the manufacturer's instructions (MiltenyiBiotec).

**Statistical analysis.** The statistical significance of survival was determined with Gehan-Breslow-Wilcoxon test. The statistical significance of luciferase activity was determined with one-way ANOVA first and then Dunnett's test. Statistical significance of comparisons between two groups was determined with a two-tailed unpaired Student's $t$-test, and $p$-values of less than 0.05 were considered to be statistically significant.

**Reporting summary.** Further information on research design is available in the Nature Research Reporting Summary linked to this article.

## Data availability
The authors declare that all data supporting the findings of this study are available within the paper and its supplementary information files. Source data are provided with this paper.

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

## Acknowledgements
This work is supported by grants from The National Key Research and Development Program of China (2016YFA0502201, 2018YFC1002801), National Natural Science Foundation of China (81373153, 81571550, 81771698, 81671613, and 31570914), Shanghai Key Laboratory of Cell Engineering (14DZ2272300), Research foundation of

Shanghai science and Technology Commission (17XD1424300), and Shanghai Leading Talents Programs.

## Author contributions

P.G., H.Y., and H.A. designed the project; P.G., X.M., M.Y., Y.Y., M.W., W.J., L.Z., Z.T., G.L., Q.Y., J.X., R.J., R.Y., S.X., M.Y., and J.P. performed the experimental work; P.G., H.A., and H.Y. analyzed results and wrote the manuscript.

## Competing interests

The authors declare no competing interests.
