## [Peer Review File. · Nature Communications]

Reviewers' comments:

Reviewer #1 (Remarks to the Author):

In this work Gao et al have examined the role of Nedd4l in innate antiviral immunity. Based on the data presented, the authors conclude that Nedd4l catalyzes cysteine ubiquitination of TRAF3 to promote antiviral immunity. The work is well designed, the data are generally clear. The main finding in the work is novel, and will be of broad interest in the field. My main criticism is the narrow focus on TRAF3. I think the authors are forcing the conclusion through that Nedd4l acts on TRAF3 in order to reach a clear model.

1. TRAF3 is mainly linked to IFN signaling downstream of RLRs and TLR3. However, the authors also observe impaired NFkB-dependent gene expression in the KO cells. This suggests a broader role for Nedd4l in PRR signaling. This needs to be addressed, and the conclusions/models needs to be modified accordingly.
2. Poly(dA:dT) activates the cGAS-STING pathway. Is TRAF3 involved in this pathway? The authors should compare the IFN/TNF induction phenotypes in Nedd4l KO, TRAF3 KO, Nedd4l, TRAF3 DKO cells after stimulation.
3. Are other TRAF proteins also ubiquitinated by Nedd4l?

Reviewer #2 (Remarks to the Author):

This paper by Gao et al. describes the identification of Nedd41 as a positive regulator of antiviral innate immunity. Nedd41 is an HECT ubiquitin E3 ligase that has not previously been implicated in innate immune regulation. By several mechanisms, the authors found that loss of Nedd41 results in less innate immune signaling to both type I IFNs and proinflammatory cytokines. They then show that Nedd41 interacts with TRAF3 and mediates the K29-linked ubiquitination of two cysteine residues of TRAF3, suggesting that this ubiquitination of TRAF3 should promote its role in positively regulating innate immunity. The finding that TRAF3 undergoes K29-linked ubiquitination on cysteines is interesting and novel because the functions of either K29-linked or cysteine-linked ubiquitin are generally unknown. While the results presented in the first part of this manuscript (up until the last figure) are fairly clear, the results become confusing in the last figure of the manuscript in which the authors attempt to prove their model. To do this, the authors made mutations in TRAF3 that prevent ubiquitination by Nedd41. Based on their model, preventing TRAF3 ubiquitination by Nedd41 should result in less innate immune signaling to type I IFNs, as Nedd41 is a positive regulator of this signaling. However, instead the authors found that blocking TRAF3 ubiquitination resulted in increased signaling to type I IFN. Therefore, it is still fundamentally unclear how Nedd41 ubiquitination of TRAF3 is actually working – is it pro-IFN or anti-IFN?

Major comments:

- If loss of Nedd41 prevents IFN-induction in response to virus, and Nedd41 ubiquitinates TRAF3 to make this happen, then how can preventing TRAF3 ubiquitination result in increased IFN-induction in response to virus? The data presented in the manuscript at this time does not support any particular model, and so it appears that the function of K29, cysteine ubiquitination on TRAF3 and the role of Nedd41 in this process is much more complex, requiring further experimentation to figure out what's going on in this system.
- Further, besides that, the ubiquitination of TRAF3 by Nedd41 appears even more complex in other ways. For example, the data presented in Fig. 5 seem to show that TRAF3 is also possibly ubiquitinated by K11-linked and/or K33-linked ubiquitin chains by Nedd41. What is the role of

these ubiquitin chains in mediating TRAF3 function? Why do the authors only focus on K29-linked ub? Could these also be important?

- Throughout the manuscript, the authors state that TRAF3 may ubiquitinate itself (see especially Fig. 5C, where TRAF3 has enhanced K63 and K48-linked ubiquitination). However, this conclusion seems to be overstated, as other endogenous E3 ligases could also be catalyzing this ubiquitination. For the authors to prove this point, they would need to use catalytic dead mutant TRAF3-C68A/H70A in these experiments.

Minor Comments:

- Labels within the figures should be more consistent. Ex) Nedd41-myc vs Myc-Nedd41 in Figure 3.
- Immunoblots should contain molecular weight markers and the differences from the 3 independent experiments should be quantified and shown in bar graphs next to the immunoblots to present the variability within the data.
- In many cases, the figure legends need more detail to describe the data presented. For example, (1) how was the RT-qPCR normalized, (2) In Sup. Fig. 4, it would be helpful if the Nedd41 domains were defined so that the reader can tell what functions are lost with the truncations, (3) the stimulation used in Fig. 7b to induce signaling to NF- κ B and IRF3 promoters is not listed.

Reviewer #3 (Remarks to the Author):

This manuscript, "E3 ligase Nedd4l promotes antiviral innate immunity by catalyzing K29-linked cysteine ubiquitination of TRAF3" by Peng Gao et al. described a new function of HECT E3 ubiquitin ligase NEDD41 in antiviral innate immunity. Utilizing a genetic approach, the authors show that conditional knockout mice lacking Nedd41 in macrophage as well as spontaneous Nedd41 deficient mice display impaired antiviral innate immune response with lower survival, less proinflammatory cytokine and type I interferon production. Through biochemical assay, the authors further demonstrate Nedd41 deficiency inhibits virus-induced TRAF3 ubiquitination, TRAF3/TBK1 complex formation and IRF3 phosphorylation, indicating TRAF3 is the target of Nedd41. This work is of significant interest as Nedd41 promote PRR-triggered K63- and K48-linked ubiquitination of lysine residue by catalyzing K29-linked ubiquitination of cysteine residue in TRAF3. Although lysine ubiquitination in PRR signaling has been intensively studied, cysteine ubiquitination in antiviral innate signaling has not been reported. Interestingly, in this manuscript, authors identify the cysteine ubiquitination of TRAF3 by Nedd41 plays an important role in antiviral innate immunity. Nevertheless, the authors should consider the following questions or concerns related to the Nedd41-TRAF3 signaling axis to strengthen their conclusions for a role of Nedd41 in antiviral innate immunity:

1. "Figure 2. Conditional Nedd4l deficiency inhibited antiviral innate immunity in vivo". Because the spontaneous Nedd41 deficient mice display reduced body weights compared to WT mice, authors explore the in vivo role of Nedd41 by generating Nedd41 conditional knockout mice. Although authors show the efficiency of conditional knockout in macrophage, more information of specificity of Nedd41^{fp/fp-cre2+/-} should be provided. Furthermore, whether Nedd41 deficiency affects macrophage development in vivo? Macrophage counts and macrophage differentiation should be considered. In Fig2.a, impairment of IFN- β and IL-6 production is consistent to that of in vitro, but why there is no difference of TNF α ?

2. "Figure 3 Nedd4l promotes TRAF3-dependent signaling.". Authors show the significant differences of proinflammatory cytokines between wild type and Nedd41 knockout macrophage, but why there is no difference of pho-p65? Since VSV infection induced proinflammatory cytokine production is mainly mediated by NF- κ B signal, authors should detect pho-IK β a and IK β a degradation. In discussion, authors suggest that c-Rel plays critical role in mediating proinflammatory cytokine induction and is inhibited by TRAF3, but they only provide the c-rel data upon LPS stimulation, they should add c-rel expression in Fig2.a. Moreover, Paul N. Moynagh et al reported that c-rel negatively regulate TLR3-mediated IFN- β production, authors need to discuss this paper.

3. "Figure 5 Nedd4l catalyzed K29-linked ubiquitination of TRAF3." Authors hypothesize that Nedd41-catalyzed K29-linked ubiquitination might facilitate K48- and K63-linked self-ubiquitination of TRAF3, but why no increase of K48- and K63-linked ubiquitination was observed in Fig5.a when overexpression of Nedd41? In Fig5.e, Ub-K no-HA may be a good control, but not the Ub-HA.
4. "Nedd4l promotes type I interferon and proinflammatory cytokine production in antiviral innate immunity by catalyzing K29-linked ubiquitination of cysteine residues in TRAF3". In Fig7., TRAF3 C56R or C124R are loss of K29-linked ubiquitination, why they could enhance the type-I interferon signaling? In Fig6., TRAF3 C56R/C124R mutation undergoes more K48- and K63-linked ubiquitination, but this is contradictory to their conclusion of Nedd41-mediated K29-linked ubiquitination promoting PRR-triggered K48-and K63-linked TRAF3 ubiquitination. Furthermore, authors do not provide enough data supporting this conclusion.
5. In Fig7., authors use luciferase assay to investigate the role of C56 and C124 of TRAF3 on IRF3 and NF-kB activation. All of these assays are explored in artificial overexpressed system, authors should confirm the endogenous activity of these two sites of TRAF3.
6. Is there any commercial antibody against K29-linked ubiquitination? If so, authors need to confirm this modification of TRAF3 upon VSV or LPS.

Reviewer #1 (Remarks to the Author):

In this work Gao et al have examined the role of Nedd4l in innate antiviral immunity. Based on the data presented, the authors conclude that Nedd4l catalyzes cysteine ubiquitination of TRAF3 to promote antiviral immunity. The work is well designed, the data are generally clear. The main finding in the work is novel, and will be of broad interest in the field. My main criticism is the narrow focus on TRAF3. I think the authors are forcing the conclusion through that Nedd4l acts on TRAF3 in order to reach a clear model.

1. TRAF3 is mainly linked to IFN signaling downstream of RLRs and TLR3. However, the authors also observe impaired NFkB-dependent gene expression in the KO cells. This suggests a broader role for Nedd4l in PRR signaling. This needs to be addressed, and the conclusions/models needs to be modified accordingly.

Response:

According to the comments, we investigated whether Nedd4l also regulates TLR2 and TLR3 signaling. As shown in Supplementary Fig.1f and 1g, Nedd4l deficiency reduces IFN- β , IL-6 and TNF- α mRNA expression in macrophages stimulated with poly(I:C). However, Nedd4l deficiency does not affect IL-6 and TNF- α mRNA expression induced by LTA. These results demonstrate Nedd4l differentially regulates innate immunity triggered by the TLR family members.

TRAF3 inhibits LPS-induced proinflammatory cytokine production by reducing IRF5 and c-Rel expression. Nedd4l deficiency decreases c-Rel expression in LPS-stimulated macrophages (Supplementary Fig. 3a). However, neither IRF5 nor c-Rel expression is affected by Nedd4l deficiency in VSV-infected macrophages (Supplementary Fig. 3b and data not shown). Furthermore, c-Rel deficiency selectively decreases IL-12p40 expression without impairing TNF- α and IL-6

production (Sanjabi S, et al. PNAS.2000;97:12705-10). According to these results and reports, it is possible Nedd4l increases LPS and VSV induced proinflammatory cytokine production by modulating other molecules rather than c-Rel and IRF5. The exact mechanism by which Nedd4l regulates RLR and TLR activated proinflammatory cytokine production needs illumination in further investigation. We modified our conclusion and discussed about this question in Page 7 and Page 14.

2. Poly(dA:dT) activates the cGAS-STING pathway. Is TRAF3 involved in this pathway? The authors should compare the IFN/TNF induction phenotypes in Nedd4l KO, TRAF3 KO, Nedd4l, TRAF3 DKO cells after stimulation.

Response:

We do not have TRAF3 KO mice. It will take too much time to introduce TRAF KO mice from elsewhere to generate Nedd4l/TRAF3 DKO cells. However, it has been reported that TRAF3 deficiency increases STING-dependent type I interferon production. TRAF3 degrades NIK, which interacts with STING to enhance IFN induction (Parvatiyar K. et al. Nat Commun. 2018 Jul 17;9(1):2770).

Nedd4l increases RLR and TLR activated IFN/TNF induction by changing the interaction of TRAF3 with other proteins rather than simply regulating TRAF3 expression. It is possible that Nedd4l increases poly(dA:dT)-induced IFN/TNF production through regulating the interaction between TRAF3 and NIK. However, it is also possible that Nedd4l increases poly(dA:dT)-induced IFN/TNF production through TRAF3-independent pathway. The comment is instructive for illuminating the mechanism by which Nedd4l regulates poly(dA:dT)-induced IFN/TNF induction. Since the present study is focused on the mechanism by which Nedd4l promotes RNA virus induced IFN/TNF production, we discuss the possibilities in Page 15 and will investigate exactly how Nedd4l promotes poly(dA:dT) and DNA virus induced IFN/TNF production in our future study.

3. Are other TRAF proteins also ubiquitinated by Nedd4l?

Response:

As shown in Supplementary Fig. 4b, Nedd41 deficiency does not reduce LPS-induced TRAF2 ubiquitination. We failed to detect the effect of Nedd41 deficiency on TRAF6 ubiquitination, because the TRAF6 antibody we used was not efficient in IP endogenous TRAF6. In some studies, TRAF6 could be immunoprecipitated with polyclonal TRAF6 antibody (Santa Cruz Biotechnology, sc-7221). Unfortunately, the company has stopped selling the product.

Reviewer #2 (Remarks to the Author):

This paper by Gao et al. describes the identification of Nedd41 as a positive regulator of antiviral innate immunity. Nedd41 is an HECT ubiquitin E3 ligase that has not previously been implicated in innate immune regulation. By several mechanisms, the authors found that loss of Nedd41 results in less innate immune signaling to both type I IFNs and proinflammatory cytokines. They then show that Nedd41 interacts with TRAF3 and mediates the K29-linked ubiquitination of two cysteine residues of TRAF3, suggesting that this ubiquitination of TRAF3 should promote its role in positively regulating innate immunity. The finding that TRAF3 undergoes K29-linked ubiquitination on cysteines is interesting and novel because the functions of either K29-linked or cysteine-linked ubiquitin are generally unknown. While the results presented in the first part of this manuscript (up until the last figure) are fairly clear, the results become confusing in the last figure of the manuscript in which the authors attempt to prove their model. To do this, the authors made mutations in TRAF3 that prevent ubiquitination by Nedd41. Based on their model, preventing TRAF3 ubiquitination by Nedd41 should result in less innate immune signaling to type I IFNs, as Nedd41 is a positive regulator of this signaling. However, instead the authors found that blocking TRAF3 ubiquitination resulted in increased signaling to type I IFN. Therefore, it is still fundamentally unclear how Nedd41 ubiquitination of TRAF3 is actually working – is it pro-IFN or anti-IFN?

Major comments:

1. If loss of Nedd41 prevents IFN-induction in response to virus, and Nedd41 ubiquitinates TRAF3 to make this happen, then how can preventing TRAF3 ubiquitination result in

increased IFN-induction in response to virus? The data presented in the manuscript at this time does not support any particular model, and so it appears that the function of K29, cysteine ubiquitination on TRAF3 and the role of Nedd41 in this process is much more complex, requiring further experimentation to figure out what's going on in this system.

Response:

In this study we find that Nedd41 directly catalyzes K29-linked ubiquitination of TRAF3 C56 and C124, and indirectly promotes K63- and K48-linked ubiquitination of TRAF3 (Fig. 5). As commented above, the results presented in the first part of the manuscript (up until the last figure) are fairly clear, the results become confusing in the last figure of the manuscript, in which C56R and C124R mutations, which prevent K29-linked ubiquitination by Nedd41 (Fig. 6a), result in increased IFN-induction in response to virus (Fig. 7e). We agree with the comments. However, it should be noted that C56R and C124R mutation prevent K29-linked TRAF3 ubiquitination, but increase K48/K63-linked TRAF3 ubiquitination.

According to the comment, we performed further experimentation to figure out what's going on in this system. To answer the question how can C56R and C124R mutation result in increased IFN-induction, the key is to explain how can C56R and C124R mutation increase K48/K63-linked ubiquitination of TRAF3 (Fig. 6b). As shown in Supplementary Fig. 6e, C56 and C124 constitute RZ1 and Z1 zinc fingers with several other cysteine and histidine residues. We propose that the conformation of the RZ1 and Z1 zinc fingers may play important roles in regulating TRAF3 ubiquitination. Nedd41-catalyzed K29-linked ubiquitination at C56 and C124 affects conformation of the zinc fingers, and the changes of the zinc fingers induced by Nedd41-catalyzed ubiquitination facilitates the interaction between TRAF3 and other E3 ligases, resulting in increased K48/K63-linked ubiquitination of TRAF3. Supporting the hypothesis, Nedd41 increases the interaction between TRAF3 with E3 ligases cIAP1/2 and HECDT3 (Fig. 7a,b), which have been proven to ubiquitinate TRAF3 by others. It is reasonable to imagine that C56R and C124R mutation affects formation of RZ1 and Z1 zinc fingers. Consistent with the model above, C56R and C124R mutation

promote interaction between TRAF3 and the E3 ligases (Fig. 7c,d), resulting in increased K48/K63-linked ubiquitination of TRAF3 (Fig. 6b) and IFN-induction in response to virus (Fig. 7e). If the model works, mutation of the other cysteine residue in RZ1 and Z1 zinc fingers will also increase K48/K63-linked ubiquitination. In fact, supporting the model, mutation of any of the cysteine and histidine residues that constitute the zinc fingers greatly increases K48/K63-linked ubiquitination as well as total ubiquitination of TRAF3 (Fig. 6d). In contrast, mutation of C61, which is nearby C56 but is not required for the formation of the zinc fingers, does not affect VSV-induced type I interferon signaling (Fig. 7e). These results provide strong evidences that RZ1 and Z1 zinc fingers play important roles in regulating TRAF3 ubiquitination. We hope we have explained clearly how can C56R and C124R mutation prevent K29-linked TRAF3 ubiquitination but increase K48/K63-linked TRAF3 ubiquitination and IFN-induction in response to virus in this system.

2. Further, besides that, the ubiquitination of TRAF3 by Nedd41 appears even more complex in other ways. For example, the data presented in Fig. 5 seem to show that TRAF3 is also possibly ubiquitinated by K11-linked and/or K33-linked ubiquitin chains by Nedd41. What is the role of these ubiquitin chains in mediating TRAF3 function? Why do the authors only focus on K29-linked ub? Could these also be important?

Response:

We only focus on K29-linked ubiquitination because Ub-K11R/K33R and Ub-K11/K33-only mutants did not consistently affect TRAF3 ubiquitination. In previously presented Fig. 5d, K11R mutation only slightly reduced TRAF3 ubiquitination, K33R mutation even increased TRAF3 ubiquitination. In previously presented Fig. 5e, Ub-K11-only and Ub-K33-only were much less efficient in TRAF3 ubiquitination compared with wild type Ub and Ub-K29-only. However, we have repeated the experiments, and the results confirm that only K29-linked ub is important (current Fig. 5e, f).

3. Throughout the manuscript, the authors state that TRAF3 may ubiquitinate itself (see especially Fig. 6C, where TRAF3 has enhanced K63 and K48-linked ubiquitination). However, this conclusion seems to be overstated, as other endogenous E3 ligases could also be catalyzing this ubiquitination. For the authors to prove this point, they would need to use catalytic dead mutant TRAF3-C68A/H70A in these experiments.

Response:

The comment is insightful. As shown in the revised Fig. 6c, C68A/H70A mutation does not affect the enhanced K48 and K63-linked ubiquitination of TRAF3(C56R) and TRAF3(C124R). This result promotes us to explore whether other endogenous E3 ligases are involved. While Nedd4l enhances interaction between TRAF3 and cIAP/HECTD3, C56R and C124R mutation also increase the interaction between TRAF3 and the E3 ligases, demonstrating that other endogenous E3 ligases are catalyzing this ubiquitination. We have revised the manuscript and concluded that other endogenous E3 ligases such as cIAP/HECTD3 catalyze the ubiquitination rather than TRAF3 ubiquitinates itself.

Minor Comments:

4. Labels within the figures should be more consistent. Ex) Nedd41-myc vs Myc-Nedd41 in Figure 3.

Response:

We have revised the labels to keep them consistent.

5. Immunoblots should contain molecular weight markers and the differences from the 3 independent experiments should be quantified and shown in bar graphs next to the immunoblots to present the variability within the data.

Response:

Molecular weight markers are contained in all of the immunoblots figures in the revised manuscript. The manuscript contains too many immunoblots. It is difficult to quantify all of the immunoblots and the figures will be too large. We quantify

p-TBK1 and p-IRF3 signals in three independent experiments, and average of the quantified signals are presented in bar graphs in Fig. 3a and Supplementary Fig. 3a. We believe that these immunoblots are the most important ones in the manuscript to demonstrate the role of Nedd4l in RLR signaling. We provide raw data of the three independent experiments in supplementary source data.

6. In many cases, the figure legends need more detail to describe the data presented. For example, (1) how was the RT-qPCR normalized, (2) In Sup. Fig. 4, it would be helpful if the Nedd41 domains were defined so that the reader can tell what functions are lost with the truncations, (3) the stimulation used in Fig. 7b to induce signaling to NF- κ B and IRF3 promoters is not listed.

Response:

According to the comments, we have revised the “Materials and methods” and the figure legends, to give more detail to describe the data presented.

Reviewer #3 (Remarks to the Author):

This manuscript, “E3 ligase Nedd4l promotes antiviral innate immunity by catalyzing K29-linked cysteine ubiquitination of TRAF3” by Peng Gao et al. described a new function of HECT E3 ubiquitin ligase NEDD41 in antiviral innate immunity. Utilizing a genetic approach, the authors show that conditional knockout mice lacking Nedd41 in macrophage as well as spontaneous Nedd41 deficient mice display impaired antiviral innate immune response with lower survival, less proinflammatory cytokine and type I interferon production. Through biochemical assay, the authors further demonstrate Nedd41 deficiency inhibits virus-induced TRAF3 ubiquitination, TRAF3/TBK1 complex formation and IRF3 phosphorylation, indicating TRAF3 is the target of Nedd41. This work is of significant interest as Nedd41 promote PRR-triggered K63- and K48-linked ubiquitination of lysine residue by catalyzing K29-linked ubiquitination of cysteine residue in TRAF3. Although lysine ubiquitination in PRR signaling has been intensively studied, cysteine ubiquitination in antiviral innate signaling has not been reported. Interestingly, in this manuscript, authors identify the cysteine ubiquitination of

TRAF3 by Nedd41 plays an important role in antiviral innate immunity. Nevertheless, the authors should consider the following questions or concerns related to the Nedd41-TRAF3 signaling axis to strengthen their conclusions for a role of Nedd41 in antiviral innate immunity:

1. "Figure 2. Conditional Nedd41 deficiency inhibited antiviral innate immunity *in vivo*". Because the spontaneous Nedd41 deficient mice display reduced body weights compared to WT mice, authors explore the *in vivo* role of Nedd41 by generating Nedd41 conditional knockout mice. Although authors show the efficiency of conditional knockout in macrophage, more information of specificity of Nedd41^{fp/fp-cre2+/-} should be provided. Furthermore, whether Nedd41 deficiency affects macrophage development *in vivo*? Macrophage counts and macrophage differentiation should be considered. In Fig2.a, impairment of IFN- β and IL-6 production is consistent to that of *in vitro*, but why there is no difference of TNF α ?

Response:

According to the comments, we detected the expression of Nedd41 in T/B cells from WT and CKO mice. We also counted the numbers of T/B cells and macrophages in spleen, bone marrow and blood. As shown in Supplementary Fig. 1c, Nedd41 CKO does not affect Nedd41 expression in T/B cells. Nedd41 CKO does not affect the number and differentiation of macrophage (Supplementary Fig. 1d).

We repeated the *in vivo* experiment. Consistent with previous experiments, Nedd41 CKO impairs IFN- β and IL-6 production, but there is no difference of TNF- α . We can't explain exactly why there is no difference of TNF- α . However, regulation of TNF- α production *in vivo* might be more complex than *in vitro*.

2. "Figure 3 Nedd41 promotes TRAF3-dependent signaling.". Authors show the significant differences of proinflammatory cytokines between wild type and Nedd41 knockout macrophage, but why there is no difference of p65? Since VSV infection induced proinflammatory cytokine production is mainly mediated by NF- κ B signal, authors should detect p65 and I κ B α degradation. In discussion, authors suggest that c-Rel plays critical role in mediating proinflammatory cytokine induction and is inhibited by TRAF3, but

they only provide the c-rel data upon LPS stimulation, they should add c-rel expression in Fig2.a. Moreover, Paul N. Moynagh et.al reported that c-rel negatively regulate TLR3-mediated IFN-beta production, authors need to discuss this paper.

Response:

According to the comments, we detected $\text{pho-I}\kappa\text{B}\alpha$ and $\text{I}\kappa\text{B}\alpha$ degradation. As shown in Fig. 3a, there is no difference of $\text{pho-I}\kappa\text{B}\alpha$ and $\text{I}\kappa\text{B}\alpha$ signals. TRAF3 is reported to inhibit LPS-induced IL-6 and TNF- α production by repressing c-Rel and IRF5 expression. Nedd4l prevents the decrease of c-Rel expression upon LPS stimulation but does not affect IRF5 expression (Supplementary Fig. 3a and data not shown). Neither c-Rel expression nor IRF5 expression is affected by Nedd4l deficiency upon VSV infection (Supplementary Fig. 3b and data not shown). During the revision of manuscript, we found it had been reported that c-Rel deficiency selectively decreased IL-12p40 expression but not TNF- α and IL-6 production (Sanjabi S, et al. PNAS.2000;97:12705-10). It seems unlikely that Nedd4l increases LPS and VSV induced IL-6 and TNF- α production via regulating c-Rel and IRF5 expression. The exact mechanism by which Nedd4l regulates proinflammatory cytokine production needs illumination in further investigation. We revise the conclusion and discussion in Page 14.

Paul N. Moynagh et.al reported that c-Rel negatively regulated TLR3-mediated IFN- β production (J Biol Chem.2011;286(52):44750-63). Overexpression of c-Rel inhibited activation of IFN- β promoter in 293 cells as shown in Fig. 4A of the report. However, overexpression of c-Rel did not decrease IFN- β production in U373 cells upon poly(I:C) stimulation as shown in Fig. 4B of the report. In a recent study, MEF cells and BMDMs expressing mutant truncated c-Rel expressed comparable level of IFN- β with wild type control MEF cells and BMDMs following HSV-1 infection (Fig. 6 in J Immunol.2019;202(5):1479-1493). The role of c-Rel in IFN- β production in innate immunity needs more investigation to confirm. We discuss the paper in Page 14 of the manuscript.

3. "Figure 5 Nedd4l catalyzed K29-linked ubiquitination of TRAF3." Authors hypothesize that Nedd41-catalyzed K29-linked ubiquitination might facilitate K48- and K63-linked self-ubiquitination of TRAF3, but why no increase of K48- and K63-linked ubiquitination was observed in Fig5.a when overexpression of Nedd41? In Fig5.e, Ub-K no-HA may be a good control, but not the Ub-HA.

Response:

In previous manuscript, the cells were not infected with virus before lysis in the overexpression experiment. In the revised manuscript, the cells are infected with virus before lysis. Increase of K48/K63-linked ubiquitination by Nedd4l are obvious as shown in Fig. 5b. These results suggest that some other signals or molecules triggered by VSV infection might be required for increase of K48/K63-linked ubiquitination of TRAF3, and Nedd41 facilitates K48/K63-linked ubiquitination of TRAF3 through an indirect mechanism.

In Fig. 5f, HA-Ub-K null is used as a control.

4. "Nedd4l promotes type I interferon and proinflammatory cytokine production in antiviral innate immunity by catalyzing K29-linked ubiquitination of cysteine residues in TRAF3". In Fig7., TRAF3 C56R or C124R are loss of K29-linked ubiquitination, why they could enhance the type-I interferon signaling? In Fig6., TRAF3 C56R/C124R mutation undergoes more K48- and K63-linked ubiquitination, but this is contradictory to their conclusion of Nedd41-mediated K29-linked ubiquitination promoting PRR-triggered K48-and K63-linked TRAF3 ubiquitination. Furthermore, authors do not provide enough data supporting this conclusion.

Response:

The questions are insightful. Since TRAF3 ubiquitination can enhance the type-I interferon signaling, the key point is how can C56R/C124R mutation, which prevent K29-linked ubiquitination, promote K48/K63-linked TRAF3 ubiquitination. As shown in Supplementary Fig. 6e, C56 and C124 constitute RZ1 and Z1 zinc fingers with several other cysteine and histidine residues. We propose that RZ1 and Z1 zinc

fingers play important roles in regulating TRAF3 ubiquitination. Nedd4l-catalyzed K29-linked ubiquitination at C56 and C124 may affect conformation of the two zinc fingers, and changes of the zinc fingers induced by Nedd4l-catalyzed ubiquitination facilitate the interaction between TRAF3 and other E3 ligases, resulting in increased K48/K63-linked ubiquitination of TRAF3. Supporting the hypothesis, Nedd4l overexpression increases the interaction between TRAF3 with cIAP1/2 and HECDT3 (Fig. 7a,b), which have been reported to ubiquitinate TRAF3. It is reasonable to imagine that C56R and C124R mutation affect formation of the two zinc fingers. Consistent with the model above, C56R and C124R mutation increase interaction between TRAF3 and the E3 ligases cIAP/HECDT3 (Fig. 7c,d), and subsequently increase K48/K63-linked ubiquitination of TRAF3 (Fig. 6b) and IFN-induction in response to virus (Fig. 7e). If the model works, mutation of the other cysteine residue in RZ1 and Z1 zinc fingers will also increase K48/K63-linked ubiquitination. Supporting the model, mutation of any of the cysteine and histidine residues that constitute the zinc fingers greatly increases K48/K63-linked ubiquitination as well as total ubiquitination of TRAF3 (Fig. 6d). In contrast, mutation of C61, which is nearby C56 but is not involved in the formation of the zinc fingers, does not increase VSV-induced type I interferon signaling (Fig. 7e). These results demonstrate that RZ1 and Z1 zinc fingers play important roles in regulating TRAF3 ubiquitination. We hope we have provided enough data supporting this conclusion.

5. In Fig7., authors use luciferase assay to investigate the role of C56 and C124 of TRAF3 on IRF3 and NF-kB activation. All of these assays are explored in artificial overexpressed system, authors should confirm the endogenous activity of these two sites of TRAF3.

Response:

To reduce artificial effect, we transfected TRAF3 KO 293T cells with optimized dose of TRAF3, TRAF3(C56R) and TRAF3(C124R) plasmid to keep expression levels of the proteins comparable with that of endogenous TRAF3 in parent wild type cells (Supplementary Fig. 7c). In these cells, mutation of C56 and C124 also significantly

increased VSV-induced IFN- β mRNA expression (Supplementary Fig. 7d), supporting the conclusion that C56 and C124 of TRAF3 are important in regulating the type-I interferon signaling.

6. Is there any commercial antibody against K29-linked ubiquitination? If so, authors need to confirm this modification of TRAF3 upon VSV or LPS.

Response:

Unfortunately, there is no commercial antibody against K29-linked ubiquitination.

REVIEWER COMMENTS

Reviewer #1 (Remarks to the Author):

This reviewer is satisfied with the response by the authors to the critics raised, and am now convinced that the conclusions are supported by the data.

Reviewer #2 (Remarks to the Author):

The authors have addressed all of my prior concerns in this revised manuscript. The new data clarifies aspects of the molecular mechanisms.

Reviewer #3 (Remarks to the Author):

In the revised manuscript, the authors have addressed most of the issues raised during the initial review. The key data supporting the conclusion of "Nedd4l promotes TRAF3 to associate with cIAP1/2 and HECTD3" were however all artificially performed by overexpression in 293T cells. Authors should try to compare endogenous TRAF3-cIAP/HECTD3 interaction in Nedd4l WT and KO cells. In addition, this conclusion cannot explain why TRAF3 differentially regulates DNA vs RNA pathways in innate immune signaling. Authors may need to discuss the possibility of additional Nedd4l targets in the DNA-induced type I interferon induction pathway. In Fig7c & d, why TRAF3(C56) level is much lower than Flag-TRAF3 when co-expression with cIAP1 or HECTD3?

Reviewer #1 (Remarks to the Author):

This reviewer is satisfied with the response by the authors to the critics raised, and am now convinced that the conclusions are supported by the data.

Response:

Thank you very much for your instructions and help to the article.

Reviewer #2 (Remarks to the Author):

The authors have addressed all of my prior concerns in this revised manuscript. The new data clarifies aspects of the molecular mechanisms.

Response:

Thank you very much for your instructions and help to the article.

Reviewer #3 (Remarks to the Author):

In the revised manuscript, the authors have addressed most of the issues raised during the initial review. The key data supporting the conclusion of "Nedd4l promotes TRAF3 to associate with cIAP1/2 and HECTD3" were however all artificially performed by overexpression in 293T cells. Authors should try to compare endogenous TRAF3-cIAP/HECTD3 interaction in Nedd4l WT and KO cells. In addition, this conclusion cannot explain why TRAF3 differentially regulates DNA vs RNA pathways in innate immune signaling. Authors may need to discuss the possibility of additional Nedd4l targets in the DNA-induced type I interferon induction pathway. In Fig7c & d, why TRAF3(C56) level is much lower than Flag-TRAF3 when co-expression with cIAP1 or HECTD3?

Response:

(1) Thank you very much for your instructions. According to the comments, we detected the interaction of TRAF3 and cIAP/HECTD3 in macrophages from Nedd4l WT and KO mice. As shown in the revised Fig. 3b, Nedd4l deficiency remarkably decreased VSV infection induced TRAF3-cIAP/HECTD3 complex formation in macrophages, providing convincing evidence that Nedd4l promoted TRAF3 to interact with cIAP1/2 and HECTD3.

(2) Previous studies demonstrated that TRAF3 promoted RNA-induced IFN production by activating TBK1. However, there was also study showing that TRAF3 negatively regulated DNA-induced IFN production by degrading NIK, which interacts with STING to enhance IFN induction. The present study demonstrates that Nedd4l positively regulates poly(I:C)-induced IFN- β production by promoting TRAF3 ubiquitination. Ubiquitination of TRAF3 by Nedd4l promotes interaction between TRAF3 with proteins such as cIAP1/2, HECTD3 and TBK1. These data do not mean ubiquitination of TRAF3 by Nedd4l will remarkably promote TRAF3-mediated NIK degradation. In this case, Nedd4l may also positively regulates poly(dA:dT)-induced IFN- β production by promoting TBK1 activation. It is worth investigating whether and how ubiquitination of TRAF3 by Nedd4l regulates NIK degradation. Besides, there is also possibility that Nedd4l regulates poly(dA:dT)-induced IFN- β production by targeting other molecules rather than TRAF3 in DNA sensing pathway. According to your comment, we discussed this issue in Page 15.

(3) We optimized purification of the plasmids and repeated the experiments in Fig7c & d. In the revised figures, Flag-TRAF3 and Flag-TRAF3(C56R) are expressed at comparable levels.

REVIEWERS' COMMENTS

Reviewer #3 (Remarks to the Author):

The Authors have now addressed all the questions I raised and revised manuscript is now suitable for publication.

Reviewer #3 (Remarks to the Author):

The authors have now addressed all the questions I raised and revised manuscript is now suitable for publication.

Response:

Thank you very much for your instructions and help to the article.